# Masked Graph Autoencoder with Non-discrete Bandwidths

## ABSTRACT

Masked graph autoencoders have emerged as a powerful graph self-supervised learning method that has yet to be fully explored. In this paper, we unveil that the existing discrete edge masking and binary link reconstruction strategies are insufficient to learn topologically informative representations, from the perspective of message propagation on graph neural networks. These limitations include blocking message flows, vulnerability to over-smoothness, and suboptimal neighborhood discriminability. Inspired by these understandings, we explore non-discrete edge masks, which are sampled from a continuous and dispersive probability distribution instead of the discrete Bernoulli distribution. These masks restrict the amount of output messages for each edge, referred to as "bandwidths". We propose a novel, informative, and effective topological masked graph autoencoder using bandwidth masking and a layer-wise bandwidth prediction objective. We demonstrate its powerful graph topological learning ability both theoretically and empirically. Our proposed framework outperforms representative baselines in both self-supervised link prediction (improving the discrete edge reconstructors by at most 20%) and node classification on numerous datasets, solely with a structure-learning pretext. Our implementation is available at https://anonymous.4open.science/r/anadnaB.

## CCS CONCEPTS

• **Computing methodologies** → **Unsupervised learning**; • **Information systems** → **Data mining**.

## KEYWORDS

Graph neural networks, graph self-supervised learning, masked graph autoencoders

**ACM Reference Format:**
Anonymous Author(s). 2024. Masked Graph Autoencoder with Non-discrete Bandwidths. In *Proceedings of Make sure to enter the correct conference title from your rights confirmation emai (Conference acronym 'XX)*. ACM, New York, NY, USA, 15 pages. https://doi.org/XXXXXXX.XXXXXXX

## 1 INTRODUCTION

Today, the demand for massive amounts of data in pre-training large models has reached an unprecedented level. *Self-supervised learning* (SSL) has emerged as a powerful approach to uncovering underlying patterns in unannotated data by pre-training on some tailor-made tasks called *pretexts* [31, 46]. Currently, SSL has

been a go-to method for pre-training large models [10, 55], gathering increasing attention. Graphs, unlike text and images, possess non-Euclidean structures that are hard for humans to intuitively understand [5] and lack well-annotated graph benchmark datasets due to the diversity of graph data and tasks. Thus, SSL is also playing a pivotal role in learning graph representations [33, 35, 75], especially in various web applications such as social recommendation [47, 73, 80].

*Taxonomy.* Contemporary graph SSL studies are mainly *contrastive methods* [68, 79, 88, 89] that leverage metric learning between augmented data pairs. In spite of this, they suffer from the thorny problem of dimensional collapse [32, 64]. On the other hand, *autoencoding methods* learn by reconstructing the input data from encoded representations. However, traditional graph autoencoders [37, 57] fall short of modeling high-dimensional representation spaces, while variational autoencoders [19, 37, 52] require additional assumptions on data distributions. In contrast, **masked graph autoencoders**, a novel framework for data reconstruction, enable the learning of high-dimensional representations without extra assumptions and show remarkable adaptability to graph data. One type of masked graph autoencoder aims to reconstruct node features, referred to as **FeatRecs** [25, 26, 83]. Another type, on which our work focuses, aims to reconstruct randomly masked links to learn graph topology, referred to as **TopoRecs** [42, 63]. Unlike language or vision tasks, TopoRecs can reconstruct links directly from graph structures without positional encoding [11, 66].

*Problem.* In this work, we focus on the **topological informativeness** of traditional TopoRecs' representations, i.e. **how well they embed graph topology into the latent representation space**. As illustrated in Figure 1(a), traditional TopoRecs rely on two key components: (i) **discrete edge masking**, where binary edge masks are sampled from a discrete distribution, and (ii) **binary link reconstruction**, which distinguishes the masked positive edges from negative ones. Despite some prominent results [42, 45, 63] derive from these two strategies, we argue that **discrete random masking and binary link reconstruction lead to limited informativeness, both globally and locally** (in Section 3.3). (i) Globally, pathways for long-range information are likely to be stretched or blocked by indiscriminate masking, leading to the vulnerability to over-smoothing. (ii) Locally, discrete masking strategies cannot provide fine-grained neighborhood discriminability, leading to suboptimal topological learning performance.

*Present work.* To tackle the problems, a topologically informative **non-discrete masking strategy** is desired. We introduce a new perspective by considering the message propagation of a GNN analogously as the (transient) transmission between nodes in a telecommunication network. By randomly setting a limit for each connection link (edge) on the amount of messages transferred in one single propagation step, named the "*bandwidth*", we manage to **mask a portion of information through each edge**, as shown in Figure 1(b). Bandwidths provide topological informativeness both globally and locally: (i) the graph topology is kept intact during

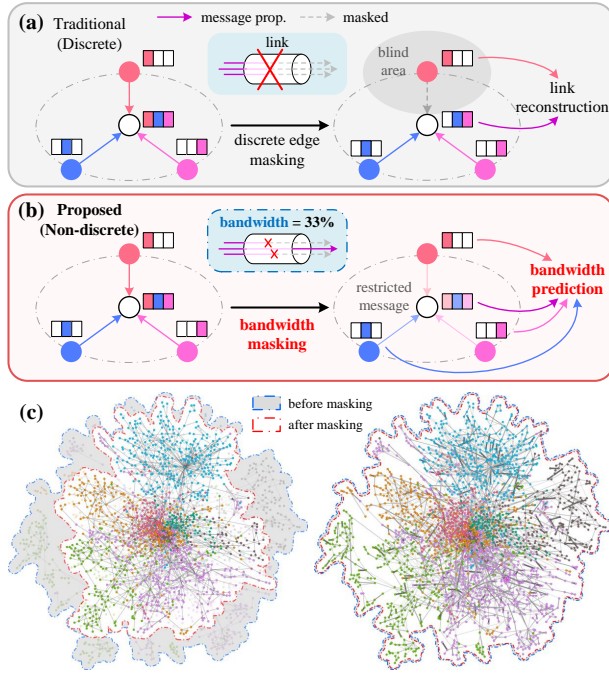

**Figure 1: Discrete masks vs. the proposed bandwidths. (a)** Traditional TopoRecs randomly mask a fixed proportion of edges and try to reconstruct them. However, messages from some neighboring nodes (e.g. the red one) as well as their predecessors will not be received by the target node (white). **(b)** We propose bandwidth masking and prediction, which first restricts the message propagated through each edge in varying degrees and then predicts how much it is restricted. The white node can now receive messages from every neighbor. **(c)** The connected component of Cora [58] before and after different masking schemes. **Left**: discretely masked graph breaks the connectivity of the original component, whereas **right**: bandwidth masked graph (where the width and grayscale of each edge denote the assigned bandwidth) keeps the original graph topology intact, so the reconstructor learns more topologically informative representations. Best viewed in color.

the training (Figure 1(c)), and long-range information can reach its destination unimpeded, which provides **global informativeness**. (ii) By sampling bandwidth values from a dispersive Boltzmann-Gibbs distribution, each node learns its neighborhood in a more fine-grained and discriminative way, which provides **local informativeness**. Accordingly, we propose a novel masked graph autoencoder with bandwidth prediction instead of link reconstruction. It is termed "Graph autoencoder aided by Bandwidths", Bandana in reverse. We showcase Bandana's great topological informativeness both theoretically (in Section 4.2) and empirically (in Section 5) by conducting extensive experiments on numerous datasets. We present the following main contributions:

- We unveil that discrete TopoRecs are insufficient to learn topologically informative representations. Globally, **blocked message flows** make the TopoRec vulnerable to the over-smoothing problem of deeper GNNs; locally, uniformized weight distribution results in the **indiscriminative neighborhood**.

- We explore a **non-discrete masking mechanism** in masked autoencoders, named "*bandwidths*". We establish a theoretical relationship between our bandwidth mechanism and regularized denoising autoencoders to prove its informativeness.
- We propose Bandana, a novel graph self-supervised learning framework that learns topologically informative representations. It outperforms representative baselines in both link prediction and node classification solely with a structure-learning pretext.

## 2 RELATED WORK

In this section, we briefly review prior work on graph self-supervised learning and compare their advantages and disadvantages.

*Contrastive Learning* [3] was born initially for the visual domain [22, 24]. The most popular contrastive learning framework [7] feeds augmented data pairs into two shared-weight neural networks. Then it computes pairwise similarities between positive and negative samples by InfoNCE contrastive loss [50]. A flood of prominent work has appeared since contrastive learning was introduced to the graph domain [74, 78, 79, 81, 85, 88, 89]. However, they must face the serious problem of representation collapse [32, 64, 72], that is, the output of the encoder will degenerate to a scalar independent of the input. In addition, InfoNCE itself does not provide the power to learn graph structures, because it measures the distance in the *feature space*. Therefore, contrastive learning methods rarely discuss the generalization performance of structure learning tasks such as link prediction unless they are specifically designed [60].

*Autoencoding*, another line of work, encodes the graph into a latent space via GNNs and then decodes the representations to reconstruct the original features or structure. The pioneering work of GAE [37] stimulated the research of traditional graph autoencoders [52, 57]. However, an overcomplete autoencoder, whose dimension of the representation space is no less than that of the data space, may degenerate into an identity map [16], severely limiting its expressive power. Graph variational autoencoders [19, 37, 45, 52] learn prior variational distributions of latent representations to obtain better latent spaces and generate new representations from them. Nevertheless, they still induce a suboptimal latent space because of the simplistic prior assumption [19].

*Mask Modeling*. The "mask-then-reconstruct" scheme has already been adopted to model natural language, computer vision, etc., and has achieved great success [10, 20, 21]. Masked autoencoders (MAEs) are simple, efficient, and almost immune to collapses, which have been introduced into the graph domain just recently [63]. As mentioned above, we call the feature space learners FeatRecs, the most well-known of which is the GraphMAE series [25, 26]. In addition, GMAE [83] introduces feature reconstruction into pre-training graph transformers. Yet, they are not suitable for link prediction due to the neglect of graph topology. TopoRecs, the structural space learners, learn by discrete random edge masks and link reconstruction, the most well-known of which are S2GAE (formerly MGAE) [63] and MaskGAE [42]. S2GAE resorts to a cross-correlation decoder to capture the information lost by perturbation, and MaskGAE employs path masks and another degree regression decoder. Despite being empirically beneficial to performance improvement, there are no theoretical guarantees that these strategies can induce a better topological learner. Moreover, despite some existing theoretical

frameworks for MAEs in the visual domain [6, 39, 51, 82], they are currently incompatible with TopoRecs. Our work aims to bridge this gap.

## 3 PRELIMINARIES

In this section, we illustrate the principle of message propagation (Section 3.1) and TopoRecs (Section 3.2). Then, we discuss the problems of discrete masking and link reconstruction (Section 3.3).

### 3.1 Notations and Concepts

We use different types of one specific symbol $S$ to denote different forms of one mathematical object. A **bold** symbol $\mathbf{S}$ denotes a matrix, with its $j$th column in **bold italics** $\boldsymbol{S}_j = \mathbf{S}_{:,j}$. The element at the $i$th row and $j$th column is in *italics* with subscripts $S_{ij}$.

Let $\mathcal{G} = (\mathbf{X}, \mathbf{A})$ be an undirected graph with $n$ nodes, where $\mathbf{X} \in \mathbb{R}^{n \times d}$ is the node feature matrix and $\mathbf{A} \in \{0, 1\}^{n \times n}$ is the adjacency matrix. We also denote by $\mathcal{E}$ the edge set and $\mathcal{V}$ the node set. $\deg(i)$ is the degree of node $i$. We call any 1-hop subgraph $\mathcal{G}_i = (\mathbf{X}^{\mathcal{G}_i}, \mathbf{A}^{\mathcal{G}_i}) \subset \mathcal{G}$ of node $i$ an *ego-graph* [87], where $\mathbf{X}^{\mathcal{G}_i} = [X_j]_{j \in \mathcal{N}_i \cup \{i\}}$, and $\mathbf{A}^{\mathcal{G}_i} \in \{0, 1\}^{n_i \times n_i}$ is a principal submatrix of $\mathbf{A}$. Here $n_i = |\mathcal{N}_i \cup \{i\}|$ with $\mathcal{N}_i$ the 1-hop neighborhood set of node $i$.

Message-passing GNNs (MPNNs) [14] learn by exchanging information between neighboring nodes. To begin with, every node receives messages from its neighbors and processes them with non-linear neurons. Then, messages are aggregated as the new representation of that node. This iterative updating mechanism can be formalized as $\mathbf{Z}^{(k)} \leftarrow \mathbf{G}\mathbf{Z}^{(k-1)}\mathbf{W}^{(k-1)}$, where $\mathbf{W}^{(k)}$ is a learnable weight matrix of the $k$th layer. For notational convenience, we integrate message aggregation and activation into one message propagation matrix $\mathbf{G}$, with the general form $\mathbf{G} = \Sigma\mathbf{A}$ where $\Sigma$ is an activation operator such as Sigmoid $\sigma(\cdot)$ and ReLU. $\mathbf{G}$ varies with GNN types, such as $\mathbf{G} = \Sigma\hat{\mathbf{D}}^{-1/2}\hat{\mathbf{A}}\hat{\mathbf{D}}^{-1/2}$ for a Graph Convolutional Network (GCN) [38]. Here $\hat{\mathbf{A}}$ represents the graph with self-loops: $\hat{\mathbf{A}} = \mathbf{A} + \mathbf{I}_n$ and $diag(\hat{\mathbf{D}}) = \mathbf{1}_n^\top\hat{\mathbf{A}}$, where $\mathbf{I}_n \in \{0, 1\}^{n \times n}$ and $\mathbf{1}_n \in \{1\}^n$ are respectively the identity matrix and the all-ones vector. To sum up, we can denote the output of the $K$th MPNN layer as $\mathbf{Z}^{(K)} = \mathbf{\Gamma}\mathbf{X}\mathbf{\Theta}$, where $\mathbf{\Gamma} = \mathbf{G}^K$ and $\mathbf{\Theta} = \prod_{i=0}^{K-1} \mathbf{W}^{(i)}$.

### 3.2 TopoRec

TopoRecs mask a subset of edges $\mathcal{E}_m \subset \mathcal{E}$ and use the unmasked set $\bar{\mathcal{E}}_m = \mathcal{E} - \mathcal{E}_m$ along with the entire node set to train an encoder. $\bar{\mathcal{E}}_m$ (resp. $\mathcal{E}_m$) induces a subgraph with adjacency matrix $\bar{\mathbf{A}}_m$ (resp. $\mathbf{A}_m = \mathbf{A} - \bar{\mathbf{A}}_m$). The masking process is defined as

$$\bar{\mathbf{A}}_m = \mathbf{A} \circ \mathbf{M}, \quad \mathbf{M} := [\mathbb{1}_{[(i,j) \in \bar{\mathcal{E}}_m]}]^{n \times n} \in \{0, 1\}^{n \times n} \quad (1)$$

with $\circ$ the Hadamard product. For discrete TopoRecs, every entry of the masking matrix $\mathbf{M}$ is an indicator $\mathbb{1}_{[(i,j) \in \bar{\mathcal{E}}_m]}$ that determines if the edge $(i, j)$ is retained. It follows an i.i.d. Bernoulli distribution $M_{ij} \sim Bernoulli(1 - p), \forall i, j$ where $p$ controls the mask ratio.

Formally, A TopoRec is a parameterized binary map $r(\mathbf{X}, \bar{\mathbf{A}}_m)$ : $\mathbb{R}^{n \times d} \times \{0, 1\}^{n \times n} \rightarrow (0, 1)^{n \times n}$ implemented by two head-to-tail networks, the so-called *encoder-decoder*. Encoder, the GNN to be pre-trained, encodes the input features as latent representations, with the $k$th-layer weights $\mathbf{W}_e^{(k)}$. Decoder plays an auxiliary role in recovering the representations, with the $k$th-layer weights $\mathbf{W}_d^{(k)}$.

A TopoRec with a single-layer MLP decoder is formalized as

$$r(\mathbf{X}, \bar{\mathbf{A}}_m) := \underbrace{\Sigma_d(b_d + \mathbf{W}_d}_{\text{decoder}} \underbrace{(\mathbf{\Gamma}\mathbf{X}\mathbf{\Theta})}_{\text{encoder}}), \ \mathbf{\Gamma} = (\Sigma_e\bar{\mathbf{A}}_m)^K, \mathbf{\Theta} = \prod_{i=0}^{K-1}\mathbf{W}_e^{(i)} \quad (2)$$

where $\Sigma_e$ (resp. $\Sigma_d$) is the activation operator of the encoder (resp. decoder). Finally, the reconstruction loss $\mathcal{L} = \mathcal{L}(r(\mathbf{X}, \bar{\mathbf{A}}_m), \mathbf{A}_m)$ minimizes the error between the output and the masked data. The cross-entropy is widely adopted to reconstruct links:

$$\mathcal{L}(r(\mathbf{X}, \bar{\mathbf{A}}_m), \mathbf{A}_m) := \mathbf{1}_n^\top \left( \frac{\delta_{\mathcal{E}_m}}{|\mathcal{E}_m|} + \frac{\delta_{\mathcal{E}^-}}{|\mathcal{E}^-|} \right)(-\mathbf{A}_m \circ \log r(\mathbf{X}, \bar{\mathbf{A}}_m))\mathbf{1}_n \quad (3)$$

Here $\delta_{\mathcal{E}} = \delta_{(i,j)}(\mathcal{E}) := \begin{cases} 1, (i,j) \in \mathcal{E} \\ 0, (i,j) \notin \mathcal{E} \end{cases}$ is the Dirac measure: the term $\frac{\delta_{\mathcal{E}_m}}{|\mathcal{E}_m|}$ in eq. (3) filters and averages every masked edge from the cross-entropy matrix $(-\mathbf{A}_m \circ \log r(\mathbf{X}, \bar{\mathbf{A}}_m))$, while $\frac{\delta_{\mathcal{E}^-}}{|\mathcal{E}^-|}$ indicates the sampled negative edge set $\mathcal{E}^- \subset \mathcal{V} \times \mathcal{V} - \mathcal{E}$.

### 3.3 A Message Propagation View of TopoRecs

In this subsection, we revisit the message propagation to answer the question "*why are discrete TopoRecs topologically uninformative?*".

*3.3.1 Global uninformativeness: blocked message flows.* A *message flow* is the directed message pathway from a source node to a sink (target) node. It has already been an explanation tool for GNN behaviors [17]. Let us analyze the message flows of a discrete TopoRec. For an ego-graph $\mathcal{G}_i$, the central node $i$ randomly selects a subset of nodes from its neighborhood $\mathcal{N}_i$ and aggregates messages from them only. This indiscriminate selection **obstructs the message flows that may be crucial to the sink nodes**. Source nodes of these message flows are not able to transmit their messages directly to the sink nodes, resulting in a large amount of stretched or even blocked flows, which are very likely to disrupt the connectivity of the original graph, as shown in Figure 2(a) (as the mask ratios of these masked autoencoders are usually very high [21, 42]).

Moreover, we reveal that **discrete masking makes the encoder vulnerable to the over-smoothing problem**. To formalize, we introduce a commonly used metric, the *Dirichlet energy* [56, 86], to evaluate the over-smoothness of a discretely masked graph. The lower the energy, the severer the over-smoothness. Our conclusion is summarized by the following theorem.

---

THEOREM 3.1 (VULNERABILITY OF DISCRETE TopoRecs TO OVER-SMOOTHING). *Let $\mathcal{G}_i = (\mathbf{X}^{\mathcal{G}_i}, \mathbf{A}^{\mathcal{G}_i})$ be an ego-graph with $n_i \geq 2$. Assume $X_j^{\mathcal{G}_i} = X_k^{\mathcal{G}_i}$ for $\forall j, k \in \mathcal{N}_i$. Define the ego Dirichlet Energy of $\mathcal{G}_i$ as*

$$E_D(\mathcal{G}_i) := \frac{1}{n_i} \sum_{j \in \mathcal{N}_i} \|X_i^{\mathcal{G}_i} - X_j^{\mathcal{G}_i}\|^2 \quad (4)$$

*If a connected component $\mathcal{G}_{i,m}$ of $\mathcal{G}_i$ is induced by imposing masks following the i.i.d. Bernoulli $(1 - p), 0 < p \leq 1$ to $\mathbf{A}^{\mathcal{G}_i}$, then we have $E_D(\mathcal{G}_{i,m}) \leq E_D(\mathcal{G}_i)$. This inequality is an equality iff $p = 1$.*

PROOF. Please refer to Appendix A.1. □

---

Note that the Dirichlet energy of the entire graph is exactly the sum of ego Dirichlet energies over all ego-graphs. So Theorem 3.1 remarks that a discretely masked graph is more likely to be over-smoothed. As such, discrete TopoRecs obtain relatively trivial expressive power with deeper GNN layers. We have also conducted

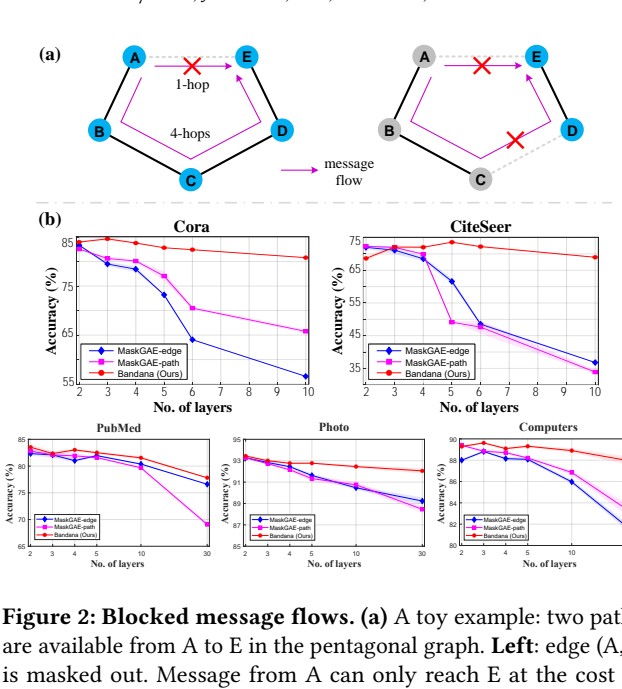

Figure 2: Blocked message flows. (a) A toy example: two paths are available from A to E in the pentagonal graph. **Left**: edge (A,E) is masked out. Message from A can only reach E at the cost of being aggregated 3 more times. **Right**: (C,D) is also masked out. E is now out of reach of A (as well as B and C). (b) Node classification accuracy of a GCN pre-trained by two counterparts of MaskGAE (blue, magenta) and Bandana (red) w.r.t. the network depth on 5 datasets (with error bands).

an experiment to verify this on 5 graph benchmark datasets. It is shown in Figure 2(b) that the performance of MaskGAE goes into a nosedive with a 5-or-more-layer GCN.

*3.3.2 Local uninformativeness: indiscriminative neighborhood.* A GCN or Graph Attention Network (GAT) [67] is usually chosen as the encoder of a TopoRec. However, both of them are **not capable of distinguishing the messages among different neighbors effectively**. We quantify the discriminability of each neighbor $j$'s message in an ego-graph $\mathcal{G}_i$ as *the dispersion of edge weights* assigned by node $i$. In GCN, such assignment is realized only by a function of node degrees $w_{ij} = 1/\sqrt{(\deg(i)+1)(\deg(j)+1)}$. This does not work well due to the power-law tendency and assortivity of networks [2, 49]. GAT, by contrast, explicitly models the neighborhood by introducing learnable self-attention matrices: $w_{ij} = \mathrm{softmax}_j(\mathrm{LeakyReLU}(\mathbf{a}^\top[\mathbf{W}_{att}Z_i||\mathbf{W}_{att}Z_j]))$, but its discriminability is limited either. Previous research [43, 44] indicates that the attention weights assigned by GAT are roughly the same for different neighbors, so effort should be made to improve the discriminative power of neighbors. Unfortunately, such power is not provided by the discrete masking and link reconstruction, because their only concern is the *existence* of links to *some* neighbors.

We have conducted another experiment on Cora for demonstration. Edge weights assigned by GCN and GAT pre-trained by MaskGAE are first computed. Then, *entropies* of the weights in every ego-graph $H_i = -\sum_{j \in \mathcal{N}_i} w_{ij} \log w_{ij}$ are calculated. The smaller the entropy value, the more discriminative the edge weights. We visualize the entropies in Figure 3 (**left**, **center**) and observe that for both GCN and GAT, the entropy distribution peaks shift towards

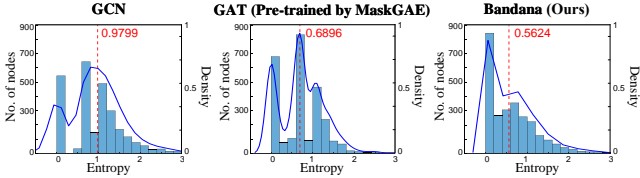

Figure 3: Entropy histograms of the edge weight distribution in ego-graphs on Cora. Blue solid lines are the Gaussian kernel density estimation curves with the red dashed lines medians.

the larger area, since MaskGAE is not able to provide discriminative power of neighbor messages for them.

To sum up, a new masking scheme is desired instead of discrete edge masking for better topological informativeness.

## 4 BANDANA

In this section, we aim to answer the questions "*what is* Bandana*?*" in Section 4.1 and "*why does* Bandana *learn informative representations?*" in Section 4.2. Bandana is a novel topological masked graph autoencoder that encompasses two main mechanisms: *continuous bandwidth masks* and *layer-wise bandwidth prediction*.

### 4.1 Bandwidth Masking and Prediction Pipeline

*4.1.1 Continuous bandwidth masks.* According to Section 3.2, a discrete TopoRec samples edge masks from a Bernoulli distribution. As bandwidths are non-discrete, each entry of the masking matrix $\mathbf{M}$ should be randomly sampled from a continuous distribution instead, with the following requirements:

(i) *Probabilistic*: $M_{ij} \in [0, 1]$ (0 for non-existent edges).
(ii) *Simplicial*: $\sum_{j \in \mathcal{N}_i} M_{ij} = 1$, $\sum_{j \notin \mathcal{N}_i} M_{ij} = 0$.
(iii) *Dispersive*: bandwidths of neighbors should be discriminative.

For (i) and (ii), every column of $\mathbf{M}$, i.e., bandwidths of neighbors in an ego-graph, should form a probabilistic simplex to stabilize the message passing process. For (iii), we want bandwidths to provide the discriminative power of neighbors. In light of these conditions, we choose the bandwidth distribution as follows.

> **Definition 4.1 (Bandwidth).** For the rest of the paper, $\mathbf{M} \in [0, 1]^{n \times n}$ is a *continuous matrix* consisting of i.i.d. probabilistic simplicial column vectors, of which each nonzero entry follows a *Boltzmann-Gibbs distribution*:
> $$M_{ij} = \mathrm{softmax}_j\left(\frac{m_{ij}}{\tau}\right) = \frac{\exp(m_{ij}/\tau)}{\sum_{k \in \mathcal{N}_j \cup \{j\}} \exp(m_{kj}/\tau)} \quad (5)$$
> where $m_{ij} \sim \mathcal{N}(0, 1)$ and $\tau$ denotes the temperature.

Intuitively, a "bandwidth" on an edge is the maximum proportion of the output messages to the input messages through that edge per message passing step. The softmax in eq. (5) plays a dual role of **normalization** and **amplification**: (i) normalization guarantees a probabilistic simplex; (ii) exponential softmax amplifies the weight dispersion in an ego-graph, which has already been discovered and utilized by some attention-based studies [54]. Though both are edge weights, bandwidths are fundamentally different from attention weights, which is further discussed in Appendix B.1.

Instead of the discrete masking matrix, we use bandwidths to perturb the adjacency matrix: $\tilde{\mathbf{A}} = \mathbf{A} \circ \mathbf{M}$. This converts the initial

discrete adjacency matrix with $A_{ij} \in \{0, 1\}$ into a continuous matrix with $\tilde{A}_{ij} \in [0, 1]$. Unlike the discrete case, both $\mathbf{M}$ and $\tilde{\mathbf{A}}$ are no longer symmetric, because there are two different bandwidth values on every edge for the bidirectional message propagation.

Note that we also adopt the temperature $\tau$ to control the bandwidth distribution. Specifically, it controls the **continuity of the mask**: when $\tau \to 0$, the Boltzmann-Gibbs distribution degenerates to the superposition of Dirac $\delta$ functions at 0 and 1, that is, the discrete Bernoulli mask; when $\tau \to \infty$, it degenerates to Uniform.

*4.1.2 Encoding.* Bandana's encoder network propagates bandwidth-restricted messages. To be more specific, the perturbed adjacency matrix $\tilde{\mathbf{A}}$ represents an undirected graph with bidirectional edge weights, on which Bandana performs message propagation instead of $\bar{\mathbf{A}}_m$. Propagation on the $k$th layer can be formalized as

$$\mathbf{Z}^{(k)} \leftarrow \tilde{\mathbf{G}}\mathbf{Z}^{(k-1)}\mathbf{W}_e^{(k-1)}, \quad \tilde{\mathbf{G}} = \Sigma_e \tilde{\mathbf{A}} \tag{6}$$

The entire encoder-decoder is defined as

$$r(\mathbf{X}, \mathbf{A}) := \underbrace{\Sigma_d(\boldsymbol{b}_d + \mathbf{W}_d}_{\text{decoder}} \underbrace{(\tilde{\boldsymbol{\Gamma}}\mathbf{X}\boldsymbol{\Theta}))}_{\text{encoder}}, \quad \tilde{\boldsymbol{\Gamma}} = \tilde{\mathbf{G}}^K \tag{7}$$

where $\Sigma_d$ refers to the softmax function. It now models the representation space in a way that every node receives and aggregates different ratios of messages from different neighbors.

*4.1.3 Bandwidth prediction.* Following the asymmetric encoder-decoder architecture [21], Bandana employs a lightweight MLP as its decoder. What the bandwidth decoder "reconstructs" is the bandwidth value of every edge, i.e. it **predicts how much every edge is masked during training**. This is in fact a logistic regression problem as Bandana aims to predict softmax probabilities, which can still be optimized by the cross-entropy objective:

$$\mathcal{L}(r(\mathbf{X}, \mathbf{A}), \tilde{\mathbf{A}}) := \mathbf{1}_n^\top \left( \frac{\delta_{\mathcal{E}}}{|\mathcal{E}|} + \frac{\delta_{\mathcal{E}^-}}{|\mathcal{E}^-|} \right) (-\tilde{\mathbf{A}} \circ \log r(\mathbf{X}, \mathbf{A}))\mathbf{1}_n \tag{8}$$

The difference is that all positive edges ($\delta_{\mathcal{E}}$) are now participating in training. We still keep the term $\delta_{\mathcal{E}^-}$ and add blocked edges as zero samples.

*4.1.4 Layer-wise masking and prediction.* It has become common knowledge that different network layers capture different granularities of information: shallower layers capture more general information, while deeper layers capture information more specific to the pretext task [27, 77, 84]. We propose a layer-wise masking scheme to explicitly capture different granularities. We generate different bandwidth masks for every layer of GNN, with the $k$th-layer perturbed adjacency matrix $\tilde{\mathbf{A}}^{(k)}$ and the corresponding message propagation matrix $\tilde{\mathbf{G}}^{(k)}$. On the backend, we share one MLP decoder for every layer. We calculate the reconstruction loss for each layer and the final loss is the average of all layer losses:

$$\mathcal{L} = \frac{1}{K}\sum_{k=0}^{K-1} \mathcal{L}^{(k)} = \frac{1}{K}\sum_{k=0}^{K-1} \mathcal{L}(\Sigma_d(\boldsymbol{b}_d + \mathbf{W}_d(\tilde{\boldsymbol{\Gamma}}^{(k)}\mathbf{X}\boldsymbol{\Theta}^{(k)})), \tilde{\mathbf{A}}^{(k)}) \tag{9}$$

where $\tilde{\boldsymbol{\Gamma}}^{(i)} = \prod_{i=0}^{k-1} \tilde{\mathbf{G}}^{(k)}, \boldsymbol{\Theta}^{(k)} = \prod_{i=0}^{k-1} \mathbf{W}_e^{(i)}$.

## 4.2 Why Are Bandwidths Informative?

In this subsection, we give empirical and theoretical support for our bandwidth schemes.

*4.2.1 A fine-grained strategy for informative topology.* The advantages of bandwidth masking and prediction are threefold.

- Compared with binary link reconstruction, predicting a continuous bandwidth value is a more **fine-grained and challenging** task which is more meaningful to the mask modeling [21].
- **Global informativeness**, as non-discrete bandwidth masks guarantee a complete graph topology and unimpeded message flows so that deeper GNNs can be pre-trained more effectively. As shown in Figure 2(b), Bandana greatly outperforms MaskGAE on GNN pre-training with 5 or more layers.
- **Local informativeness**, as Bandana can provide the discriminative power of neighbor messages by bandwidth prediction. As shown in Figure 3 (**right**), Bandana has the most discriminative neighborhood weights.

*4.2.2 Implicit optimization on the topological manifold.* Bandana's excellent topological learning ability is theoretically guaranteed. We elucidate that Bandana can be interpreted as a regularized denoising autoencoder [70] in an implicit graph topological space, while a discrete TopoRec cannot. Furthermore, bandwidth prediction is mathematically equivalent to optimizing a "score" in that space. To this end, we first assign each column of the adjacency matrix $\mathbf{A}$ to the corresponding node in the graph as its new "feature".

> *Definition 4.2 (Topological encoding).* A *topological encoding matrix* is defined as $\mathbf{T} := \mathbf{A} - \mathbf{1}_n\mathbf{1}_n^\top \in \mathbb{R}^{n \times n}$, where $\mathbf{1}_n$ is the all-ones vector. Denote $\boldsymbol{T}_j$ as the *topological encoding* of node $j$.

One advantage of the topological encoding is that it allows us to write the bandwidth masking as adding random noises on it:

$$\tilde{\mathbf{T}} = \delta_{\mathcal{E}}(\mathbf{T} + \boldsymbol{\epsilon}) = \mathbf{T} + \boldsymbol{\epsilon}, \quad \boldsymbol{\epsilon}_i \sim \text{softmax}(\mathcal{N}(0, \mathbf{I}_n)) \tag{10}$$

Assume $\boldsymbol{T}_j$ and the perturbed $\tilde{\boldsymbol{T}}_j$ follow the probability distributions $p(\boldsymbol{T}_j)$ and $p(\tilde{\boldsymbol{T}}_j)$ respectively. From the topological encoding perspective, $\mathbf{X}$ and $\boldsymbol{\Theta}$ are conversely the non-linear transformations on $\tilde{\mathbf{A}}$, in which case we denote our bandwidth reconstructor by $r_{\mathbf{X}}$. Under this premise, the following Proposition gives that $r_{\mathbf{X}}$ can be viewed as a regularized denoising autoencoder.

> PROPOSITION 4.3 (NON-DISCRETE TopoRec IS A DENOISING AUTOENCODER). *Suppose a TopoRec on vectors $r_{\mathbf{X}} : \mathbb{R}^n \to \mathbb{R}^n$ is at least first-order differentiable (to $\boldsymbol{T}_j$ for the rest of the paper). If the perturbed topological encoding $\tilde{\boldsymbol{T}}_j$ on a connected graph follows a continuous distribution $p(\tilde{\boldsymbol{T}}_j)$ and satisfies*
> 
>  *(i) $n \ll 2|\mathcal{E}|$, and*
>  *(ii) all elements in $\{\tilde{\boldsymbol{T}}_j\}_{j=1}^n$ follow an i.i.d. isotropic multivariate Gaussian $\mathcal{N}(\boldsymbol{\mu}_{\tilde{\boldsymbol{T}}_j}, \tilde{\Sigma}_{\tilde{\boldsymbol{T}}_j})$, i.e. the covariance matrix satisfies $\Sigma_{\tilde{\boldsymbol{T}}_j} = c\mathbf{I}$ where $c$ is an arbitrary constant.*
> 
> *Then, $\mathcal{L}$ in eq. (8) defines a regularized denoising autoencoder:*
> 
> $$\mathcal{L} = \underbrace{\mathbb{E}_{j \in \mathcal{V}}[\|r_{\mathbf{X}}(\boldsymbol{T}_j) - \boldsymbol{T}_j\|^2]}_{\text{reconstruction}} + \underbrace{\sigma_\epsilon^2 \mathbb{E}_{j \in \mathcal{V}}[\|\nabla r_{\mathbf{X}}(\boldsymbol{T}_j)\|_F^2]}_{\text{regularization}} + o(\sigma_\epsilon^2) \tag{11}$$
> 
> *where $\sigma_\epsilon^2$ is the noise variance.*
> 
> PROOF. Please refer to Appendix A.2. We also discuss the mildness of the assumptions in Appendix A.4. □

Proposition 4.3 is consistent with previous studies that the masked autoencoder is a kind of denoising autoencoder [21], but the noise

should technically be non-discrete for TopoRecs. According to the existing analysis of the denoising autoencoder [1], we have the following theorem.

---

Theorem 4.4 (Bandwidth prediction optimizes in the topological encoding space). *Suppose $r_X$ fulfills the condition in Proposition 4.3. Then for $\sigma_\epsilon^2 \to 0$, the optimal TopoRec $r_X^* = \arg\min_{r_X} \mathcal{L}$ is asymptotically equivalent to an implicit gradient optimizer of $\log p(T_j)$:*

$$r_X^*(T_j) - T_j \propto \nabla \log p(T_j) \qquad (12)$$

Proof. Please refer to Appendix A.3. □

---

By Theorem 4.4, the optimal bandwidth predictor ($r_X = r_X^*$) optimizes the gradient of log probabilities of the topological encoding, indicating that the bandwidth masking and prediction scheme are **theoretically learning graph topology**. Based on our conclusions, Bandana can be further expanded to some theoretically grounded frameworks, such as score-based models [61] and energy-based models [40]. We have further discussions in Appendix B.2.

## 5 EXPERIMENTS AND RESULT ANALYSES

In this section, we first introduce experimental configurations in Section 5.1. More detailed settings can be found in Appendix C. Then, we showcase the experiment results of Bandana to answer the following research questions:

- RQ1. *Is Bandana able to learn more informative topology than discrete TopoRecs in practice?*
- RQ2. *How does Bandana perform on node classification?*
- RQ3. *How does Bandana perform on link prediction?*
- RQ4. *How effective are Boltzmann-Gibbs bandwidths and the layer-wise strategy?*
- RQ5. *How does the temperature affect Bandana's performance?*

### 5.1 Experimental Settings

*Datasets.* Apart from the two synthetic datasets in Section 5.2, we conduct experiments on 9 well-known undirected and unweighted graph benchmark datasets, including (i) citation networks: Cora, CiteSeer, PubMed [58]; (ii) co-purchase networks: Photo, Computers [59]; (iii) co-author networks: CS, Physics [59]; (iv) OGB networks: ogbn-arxiv (for node classification), ogbl-collab (for link prediction) [28]. Detailed statistics can be found in Appendix C.1.

*Reproducibility.* We report all quantitative results as "mean ± standard deviation" by running 10 times under the same setup. Hardware, training setups, and hyperparameters can be found in Appendix C.2 and C.3.

*Baselines.* As self-supervised methods are being studied, only self-supervised algorithms are considered as baselines. They are divided into the following categories: (i) traditional graph autoencoders: GAE [37], ARGA [52]; (ii) variational graph autoencoders: VGAE [37], ARVGA [52], SIG-VAE [19], SeeGera [45][1]; (iii) contrastive and non-contrastive (with no negative sampling) methods: GRACE [88], GCA [89], COSTA [85], CCA-SSG [81], T-BGRL [60]; (iv) FeatRecs: GraphMAE [26], GraphMAE2 [25]; (v) TopoRecs: S2GAE [63], MaskGAE-edge, MaskGAE-path [42] (with edge masking and path masking

---

[1]SeeGera fuses mask modeling with the variational autoencoder. We still count it as variational-based in light of its generative characteristic and learning objective.

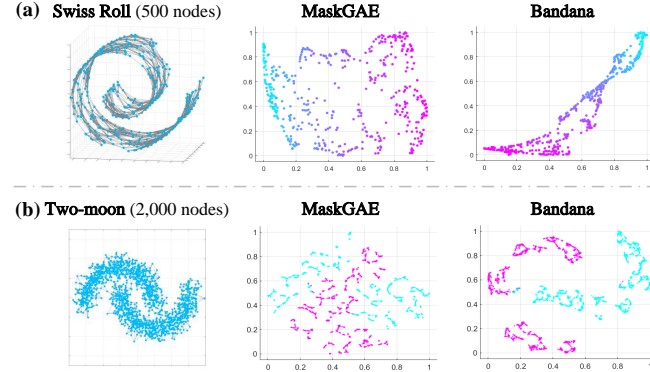

**Figure 4: Manifold learning visualizations. (a)** The Swiss Roll. MaskGAE only learns suboptimal representations loosely scattered in the latent space, whereas Bandana learns a more compact surface. **(b)** The Two-moon. While MaskGAE does not give informative results, Bandana successfully learns the crescent-shaped topology.

separately). We use "†" to mark baselines that are implemented by us for the current task because they are not officially implemented.

### 5.2 Learning Topological Manifolds (RQ1)

We verify Bandana's informative representation learning ability by performing manifold learning on two undirected synthetic datasets with structures: *Swiss Roll*, a curved surface on $\mathbb{R}^3$; and *Two-moon*, two interleaved crescent-shaped clusters on $\mathbb{R}^2$. We assign a column of the identity matrix $I_n$ to each node as features with no topological information. The latent representation spaces are learned by MaskGAE and Bandana respectively (each model is trained till the early stopping) and visualized by t-SNE [65]. It is obvious in Figure 4 that Bandana is more topologically informative than MaskGAE.

### 5.3 Comparison on Node Classification (RQ2)

Similar to other self-supervised models [21, 26, 42, 68, 81], Bandana follows the **linear probing** setup to evaluate. That is, we use the pre-trained encoder's output representations to train a Xavier-initialized [15] linear layer.

We report two classic metrics, Micro-F1 and Macro-F1, in Table 1. It is evident that Bandana achieves competitive performance with state-of-the-art contrastive methods and FeatRecs, but only with a structure-learning pretext. As indicated by the Avg. Rank, the performance of discrete TopoRecs (S2GAE, MaskGAE) in node classification tasks is difficult to emulate the dominant FeatRecs (GraphMAE, GraphMAE2). However, Bandana surpasses both settings of MaskGAE on 7/8 datasets, surpasses COSTA (one of the most advanced contrastive frameworks) on 6/7 datasets, and outperforms GraphMAE and GraphMAE2 by 1.2 and 2.8 ranks, respectively. Our work, perhaps surprisingly, shows that fine-grained topological learning can uncover the close relationship between the graph structure and the intrinsic characteristics of node features.

### 5.4 Comparison on Link Prediction (RQ3)

Unlike node classification, our evaluation of link prediction is *different from the old routine*. Previous TopoRecs directly performed link prediction in an end-to-end manner without probing or fine-tuning

**Table 1: Micro-F1(%) and Macro-F1(%) of node classification.** Best results in each column are in **bold**. "Avg. Rank" stands for the average rank. "OOM" stands for "Out-Of-Memory" on a 24GB GPU.

| Micro-F1 Macro-F1 | Year | Model | Cora | CiteSeer | PubMed | Photo | Computers | CS | Physics | ogbn-arxiv | Avg. Rank |
|---|---|---|---|---|---|---|---|---|---|---|---|
| Traditional Autoencoder | 2016 | GAE† [37] | 80.15 ± 0.34
78.44 ± 0.90 | 69.79 ± 0.36
62.64 ± 0.58 | 80.51 ± 0.53
79.62 ± 0.38 | 91.07 ± 0.09
89.83 ± 0.13 | 87.92 ± 0.12
86.02 ± 0.25 | 90.46 ± 0.29
89.75 ± 0.32 | 93.04 ± 0.03
91.02 ± 0.04 | 69.58 ± 0.32
48.25 ± 0.53 | 10.3 |
| | 2018 | ARGA† [52] | 77.93 ± 0.59
76.89 ± 0.51 | 68.55 ± 0.34
63.33 ± 0.68 | 77.78 ± 0.63
76.54 ± 0.82 | 92.77 ± 0.26
91.60 ± 0.10 | 88.11 ± 0.08
86.34 ± 0.15 | 92.46 ± 0.14
90.61 ± 0.19 | 94.32 ± 0.04
92.58 ± 0.07 | 69.81 ± 0.27
47.89 ± 0.45 | 9.3 |
| Variational Autoencoder | 2016 | VGAE† [37] | 76.30 ± 0.49
74.70 ± 0.60 | 58.85 ± 0.79
52.80 ± 0.91 | 75.73 ± 0.22
75.39 ± 0.24 | 89.58 ± 0.20
86.61 ± 0.40 | 84.99 ± 0.19
82.26 ± 0.30 | 92.33 ± 0.07
89.09 ± 0.16 | 94.40 ± 0.07
92.28 ± 0.11 | 69.94 ± 0.30
47.05 ± 0.79 | 12.3 |
| | 2018 | ARVGA† [52] | 76.85 ± 0.88
75.15 ± 0.95 | 54.73 ± 0.46
48.71 ± 1.27 | 73.06 ± 0.42
73.49 ± 0.44 | 89.51 ± 0.23
86.88 ± 0.40 | 85.03 ± 0.15
81.54 ± 0.36 | 92.56 ± 0.09
89.67 ± 0.23 | 93.64 ± 0.08
91.08 ± 0.15 | 69.39 ± 0.36
47.34 ± 0.59 | 12.5 |
| | 2023 | SeeGera [45] | 83.95 ± 0.55
82.88 ± 0.66 | 72.11 ± 1.26
68.48 ± 0.86 | 79.55 ± 0.29
78.36 ± 0.44 | 90.13 ± 0.57
87.76 ± 1.01 | 88.39 ± 0.26*
87.76 ± 1.01 | 88.79 ± 0.93
85.38 ± 1.62 | OOM | OOM | 9.6 |
| Contrastive & Non-contrastive | 2020 | GRACE [88] | 80.95 ± 0.29
79.20 ± 0.44 | 70.39 ± 0.46
68.15 ± 0.32 | 83.55 ± 0.44
83.29 ± 0.20 | 92.12 ± 0.14
90.99 ± 0.36 | 87.68 ± 0.15
85.82 ± 0.27 | 91.90 ± 0.01
89.09 ± 0.01 | OOM | OOM | 8.3 |
| | 2021 | GCA [89] | 81.92 ± 0.17
80.76 ± 0.35 | 71.60 ± 0.27
**68.79 ± 0.37** | **84.08 ± 0.16**
**83.70 ± 0.29** | 92.39 ± 0.20
91.17 ± 0.30 | 87.14 ± 0.15
85.10 ± 0.31 | 92.61 ± 0.06
90.64 ± 0.16 | OOM | OOM | 6.4 |
| | 2022 | COSTA [85] | 84.60 ± 0.20
82.50 ± 0.21 | 72.57 ± 0.31
66.11 ± 0.29 | 83.76 ± 0.03
83.16 ± 0.02 | 90.98 ± 0.00
88.22 ± 0.00 | 87.35 ± 0.08
85.99 ± 0.13 | 92.48 ± 0.05
89.32 ± 0.08 | 95.31 ± 0.04
93.90 ± 0.05 | OOM | 6.4 |
| | 2021 | CCA-SSG [81] | 83.96 ± 0.38
83.01 ± 0.48 | 73.45 ± 0.44
68.75 ± 0.51 | 81.81 ± 0.53
81.31 ± 0.51 | 92.87 ± 0.36
91.69 ± 0.49 | 88.61 ± 0.29
87.24 ± 0.52 | 93.01 ± 0.29
90.73 ± 0.51 | 95.31 ± 0.07
93.76 ± 0.10 | 69.52 ± 0.09
47.39 ± 0.51 | 5.1 |
| FEATREC | 2022 | GraphMAE [26] | 84.05 ± 0.59
**83.07 ± 0.53** | 73.06 ± 0.37
67.78 ± 0.85 | 80.98 ± 0.47
80.26 ± 0.48 | 92.92 ± 0.40
91.93 ± 0.47 | 89.24 ± 0.45
**88.12 ± 0.78** | 93.09 ± 0.14
**91.42 ± 0.15** | **95.65 ± 0.07**
94.19 ± 0.09 | 71.30 ± 0.24
**51.15 ± 0.15** | 3.6 |
| | 2023 | GraphMAE2 [25] | 83.84 ± 0.54
82.80 ± 0.46 | 73.48 ± 0.34
68.70 ± 0.42 | 81.34 ± 0.44
80.68 ± 0.43 | 93.30 ± 0.20
92.19 ± 0.24 | 89.01 ± 1.53
87.63 ± 1.79 | 91.31 ± 0.07
88.89 ± 0.13 | 95.25 ± 0.05
93.78 ± 0.07 | **71.82 ± 0.00**
50.42 ± 0.00 | 5.2 |
| TOPOREC | 2023 | S2GAE [63] | 78.34 ± 0.96
77.44 ± 0.86 | 65.31 ± 0.64
62.54 ± 0.64 | 80.11 ± 0.52
79.04 ± 0.47 | 91.43 ± 0.07
90.47 ± 0.15 | 85.31 ± 0.07
81.48 ± 0.18 | 90.47 ± 0.07
87.69 ± 0.11 | 93.98 ± 0.06
91.95 ± 0.08 | 67.77 ± 0.36
36.41 ± 0.24 | 11.8 |
| | 2023 | MaskGAE-edge [42] | 83.33 ± 0.15
82.60 ± 0.24 | 72.02 ± 0.46
66.36 ± 0.63 | 82.33 ± 0.39
81.63 ± 0.41 | 93.28 ± 0.08
92.04 ± 0.08 | 89.42 ± 0.15
88.00 ± 0.14 | 92.29 ± 0.25
90.17 ± 0.34 | 95.10 ± 0.04
93.48 ± 0.04 | 70.95 ± 0.29
49.37 ± 0.45 | 5.9 |
| | 2023 | MaskGAE-path [42] | 82.54 ± 0.16
81.84 ± 0.26 | 72.32 ± 0.39
65.77 ± 0.44 | 82.80 ± 0.22
82.23 ± 0.23 | 93.29 ± 0.10
92.16 ± 0.17 | 89.40 ± 0.10
87.69 ± 0.15 | 92.54 ± 0.21
90.25 ± 0.31 | 95.15 ± 0.11
93.51 ± 0.03 | 71.22 ± 0.40
49.99 ± 0.54 | 5.4 |
| | | Bandana | **84.62 ± 0.37**
82.97 ± 0.92 | **73.60 ± 0.16**
68.11 ± 0.48 | 83.53 ± 0.51
82.99 ± 0.40 | **93.44 ± 0.11**
**92.26 ± 0.04** | **89.62 ± 0.09**
87.79 ± 0.20 | **93.10 ± 0.05**
91.02 ± 0.13 | 95.57 ± 0.04
**94.20 ± 0.05** | 71.09 ± 0.24
49.66 ± 0.50 | **2.4** |

*We obtain a much lower score for SeeGera on Computers than the official one. We report the Micro-F1 from the original paper [45] instead.

since they do the exact same thing for pre-training. However, it is not a self-supervised case and hence not suitable for evaluating self-supervised models. Thus, we utilize a fairer evaluation scheme called **dot-product probing**, which replaces the original MLP decoder with a dot-product operator $\mathbf{A}_{\text{recon}} = \sigma(\mathbf{Z}\mathbf{Z}^{\top})$, as SeeGera does [45]. We employ the dot-product probing instead of the end-to-end training for Bandana *as well as all baselines* (note that this may lead to some discordance between our results and those officially reported).

We report Area Under the ROC curve (AUC) and Average Precision (AP) in Table 2. We have several observations. (i) Despite no longer using link prediction for pre-training, Bandana still achieves the best link prediction results. In particular, it greatly outperforms the performance of MaskGAE by 20% on Computers. (ii) Bandana gains over 3%-10% improvement compared to the best contrastive results. From the Avg. Rank, the performance of contrastive methods under dot-product probing is less than satisfactory, even for the advanced link prediction model T–BGRL, because they do not explicitly learn graph structures while pre-training. (iii) FEATRECs (GraphMAE, GraphMAE2) do not perform as well as TOPORECs and even traditional autoencoders (GAE, ARGA), since they only pay attention to node features. (iv) Some contrastive methods and variational autoencoders require more memory for large graphs. This highlights the lightweight property of TopoRecs. More experiment results of link prediction, including Hits@20/Hits@50 on ogbl-collab and further analyses of the probing setup, can be found in Appendix D.

## 5.5 Ablation and Parameter Analysis

*5.5.1 The masking strategy (RQ4).* We have analyzed the strengths of Boltzmann-Gibbs bandwidths and the layer-wise strategy in Section 4.1. To validate these strengths, we experiment with different distributions, including the discrete Bernoulli distribution $M_{ij} \sim Bernoulli(p)$, uniform distribution $M_{ij} \sim U(2p-1, 1)(p > 0.5)$, truncated Gaussian distribution $M_{ij} \sim \psi(p, 1, 2p-1, 1)(p > 0.5)^2$, and the Boltzmann-Gibbs distribution in eq. (5) (we ensure that masks sampled from these distributions have the same mask ratio $p$). These variants only feed the output-layer representations into the decoder. The model employing layer-wise masking only (each layer uses an independent mask set but only the last layer performs the prediction) is referred to as LWM, while the one with both layer-wise masking and prediction is referred to as LWP. It is obvious from Table 3 that the model with Boltzmann-Gibbs bandwidths outperforms all models with different mask distributions. Furthermore, Bandana's setting obtains the best node classification performance on all three datasets. Note that the model with LWM only learns suboptimal representations because it only attempts to predict one set of masks while multiple different sets are used.

*5.5.2 Effect of the temperature (RQ5).* As discussed in Section 4.1, the temperature $\tau$ of the Boltzmann-Gibbs distribution controls the continuity of the mask. This is also called *temperature scaling* in the

---

[2]$\psi(\mu, \sigma^2, a, b)$ denotes a Gaussian distribution $\mathcal{N}(\mu, \sigma^2)$ truncated within the interval $[a, b]$ where $-\infty < a < b < +\infty$.

**Table 2: AUC(%) and AP(%) of link prediction.** Best results in each column are in **bold**. "OOM" stands for "Out-Of-Memory" on a 24GB GPU. "NODATA" means that the model cannot perform due to the specific data format.

| AUC / AP | Year | Model | Cora | CiteSeer | PubMed | Photo | Computers | CS | Physics | Avg. Rank |
|---|---|---|---|---|---|---|---|---|---|---|
| Traditional Autoencoder | 2016 | GAE [37] | 94.66 ± 0.26
94.22 ± 0.39 | 95.19 ± 0.45
95.70 ± 0.31 | 94.58 ± 1.12
94.26 ± 1.65 | 71.45 ± 0.95
65.99 ± 0.96 | 70.99 ± 1.03
67.88 ± 0.82 | 93.78 ± 0.36
89.87 ± 0.59 | 88.88 ± 1.11
82.45 ± 1.59 | 8.6 |
|  | 2018 | ARGA [52] | 94.76 ± 0.18
94.93 ± 0.20 | 95.68 ± 0.35
96.34 ± 0.25 | 94.12 ± 0.08
94.19 ± 0.08 | 85.42 ± 0.79
80.58 ± 1.40 | 67.09 ± 3.93
62.53 ± 3.17 | 95.49 ± 0.17
92.56 ± 0.33 | 90.70 ± 1.08
89.37 ± 1.16 | 6.5 |
| Variational Autoencoder | 2016 | VGAE [37] | 91.24 ± 0.48
92.27 ± 0.43 | 94.55 ± 0.48
95.34 ± 0.37 | 95.46 ± 0.04
94.29 ± 0.07 | 95.61 ± 0.05
94.63 ± 0.06 | 92.69 ± 0.03
88.27 ± 0.08 | 87.34 ± 0.43
80.24 ± 0.55 | 89.27 ± 0.83
82.79 ± 1.14 | 6.5 |
|  | 2018 | ARVGA [52] | 91.35 ± 0.87
91.98 ± 0.85 | 94.47 ± 0.33
95.21 ± 0.33 | 96.17 ± 0.21
94.81 ± 0.41 | 95.44 ± 0.14
94.49 ± 0.12 | 92.38 ± 0.15
88.49 ± 0.33 | 87.39 ± 0.37
80.31 ± 0.49 | 88.96 ± 0.96
82.38 ± 1.31 | 6.9 |
|  | 2019 | SIG-VAE [19] | 90.36 ± 1.34
91.36 ± 1.16 | 88.85 ± 0.69
90.27 ± 0.73 | OOM | OOM | OOM | OOM | OOM | 11.0 |
|  | 2023 | SeeGera [45] | 95.49 ± 0.70
**95.90 ± 0.64** | 94.61 ± 1.05
96.40 ± 0.89 | 95.19 ± 3.94
94.60 ± 4.17 | 95.25 ± 1.19
94.04 ± 1.18 | 96.53 ± 0.16
96.33 ± 0.16 | 95.73 ± 0.70
93.17 ± 0.53 | OOM | 3.8 |
| Contrastive & Non-contrastive | 2020 | GRACE† [88] | 81.80 ± 0.45
82.02 ± 0.50 | 84.78 ± 0.38
82.85 ± 0.36 | 93.11 ± 0.37
92.88 ± 0.30 | 88.64 ± 1.17
83.85 ± 4.15 | 89.97 ± 0.25
92.15 ± 0.43 | 87.67 ± 0.10
94.87 ± 0.02 | OOM | 9.2 |
|  | 2021 | GCA† [89] | 81.91 ± 0.76
80.51 ± 0.71 | 84.72 ± 0.28
81.57 ± 0.22 | 94.33 ± 0.67
93.13 ± 0.62 | 89.61 ± 1.46
86.53 ± 3.00 | 90.67 ± 0.30
90.50 ± 0.63 | 88.05 ± 0.00
94.94 ± 0.37 | OOM | 8.7 |
|  | 2021 | CCA-SSG† [81] | 67.54 ± 1.30
72.74 ± 1.18 | 78.88 ± 2.73
77.42 ± 4.56 | 74.97 ± 0.28
77.11 ± 0.26 | 91.04 ± 2.98
89.68 ± 3.85 | 83.85 ± 1.35
84.04 ± 1.74 | 83.54 ± 0.98
78.66 ± 1.06 | 77.40 ± 0.08
73.33 ± 0.10 | 12.0 |
|  | 2023 | T-BGRL [60] | 73.18 ± 0.54
76.81 ± 0.73 | 78.11 ± 0.48
83.15 ± 0.47 | 76.21 ± 0.18
80.99 ± 0.13 | 80.80 ± 0.04
84.34 ± 0.06 | 84.60 ± 0.05
86.85 ± 0.05 | 70.08 ± 0.12
79.50 ± 0.09 | 89.18 ± 0.04
84.30 ± 0.05 | 11.1 |
| FeatRec | 2022 | GraphMAE† [26] | 93.02 ± 0.53
91.40 ± 0.59 | 95.21 ± 0.47
94.42 ± 0.67 | 87.54 ± 1.06
86.93 ± 1.01 | 75.08 ± 1.24
70.04 ± 1.12 | 71.27 ± 0.89
66.84 ± 1.10 | 92.45 ± 4.18
91.67 ± 4.17 | 85.03 ± 7.16
82.46 ± 9.33 | 9.9 |
|  | 2023 | GraphMAE2† [25] | 93.26 ± 1.00
91.65 ± 0.98 | 95.26 ± 0.14
94.36 ± 0.20 | 90.85 ± 0.91
90.37 ± 0.92 | 73.03 ± 2.24
68.77 ± 1.50 | 72.20 ± 2.09
67.97 ± 1.52 | 94.57 ± 0.32
92.76 ± 0.54 | 94.56 ± 0.81
93.86 ± 1.09 | 8.1 |
| TopoRec | 2023 | S2GAE [63] | 89.27 ± 0.33
89.78 ± 0.22 | 86.35 ± 0.42
87.38 ± 0.29 | 89.53 ± 0.23
88.68 ± 0.33 | NODATA | NODATA | NODATA | NODATA | 12.0 |
|  | 2023 | MaskGAE-edge [42] | 95.66 ± 0.16
94.65 ± 0.24 | 97.02 ± 0.27
96.89 ± 0.45 | 96.51 ± 0.82
96.08 ± 0.68 | 81.12 ± 0.45
77.11 ± 0.40 | 76.23 ± 3.13
71.71 ± 2.90 | 92.41 ± 0.44
87.16 ± 0.69 | 91.94 ± 0.37
86.33 ± 0.55 | 5.5 |
|  | 2023 | MaskGAE-path [42] | 95.47 ± 0.25
94.64 ± 0.25 | **97.21 ± 0.17**
97.02 ± 0.32 | 97.19 ± 0.18
96.69 ± 0.19 | 80.46 ± 0.34
76.56 ± 0.55 | 73.24 ± 1.26
70.94 ± 1.26 | 87.96 ± 0.44
80.84 ± 0.58 | 86.19 ± 0.36
78.55 ± 0.45 | 6.9 |
|  |  | Bandana | **95.71 ± 0.12**
95.25 ± 0.16 | 96.89 ± 0.21
**97.16 ± 0.17** | **97.26 ± 0.16**
**96.74 ± 0.38** | **97.24 ± 0.11**
**96.79 ± 0.15** | **97.33 ± 0.06**
**96.91 ± 0.09** | **97.42 ± 0.08**
**97.09 ± 0.15** | **97.02 ± 0.04**
**96.67 ± 0.05** | **1.2** |

**Table 3: Effect of the masking strategies on the average node classification accuracy (%).** Best results in each column are in **bold**. Second-best results in each column are underlined.

| Variants | Cora | CiteSeer | PubMed |
|---|---|---|---|
| Bernoulli | 79.16 ± 0.15 | 68.60 ± 0.90 | 82.67 ± 0.40 |
| Uniform | 81.36 ± 0.20 | 70.25 ± 0.55 | 81.84 ± 0.47 |
| Truncated Gaussian | 79.34 ± 0.46 | 69.95 ± 0.25 | 82.00 ± 0.56 |
| Boltzmann-Gibbs | 84.02 ± 0.09 | 72.45 ± 0.42 | 83.31 ± 0.38 |
| Boltzmann-Gibbs, LWM | 82.38 ± 0.19 | 70.75 ± 0.55 | 81.70 ± 0.57 |
| Bandana (Boltzmann-Gibbs, LWP) | **84.62 ± 0.37** | **73.60 ± 0.16** | **83.53 ± 0.51** |

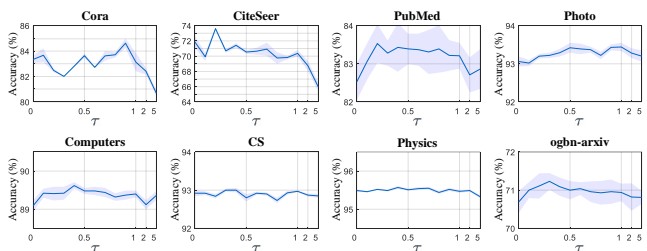

**Figure 5: Node classification accuracy w.r.t. the temperature.**

field of calibration [18], distillation [23], etc. Figure 5 illustrates the node classification performance with different values of $\tau$, set as 1e-6, 0.1, 0.2, ..., 0.9, 1, 2, and 5. The extent of performance fluctuation w.r.t. temperature varies across datasets, as does the temperature range for the best accuracy, such as [0.8, 0.9] for Cora and [0.2, 1] for PubMed. However, it can be observed on the vast majority of datasets that the model performance declines if $\tau$ is too small (i.e. the discretized mask) or too large (i.e. the uniformized mask).

# 6 CONCLUSION

This work firstly discusses two limitations in the message propagation of existing discrete TopoRecs, which induce the insufficiency of learning topologically informative representations. To address the issues, we explore non-discrete masking by a novel bandwidth masking and reconstruction scheme. We present our masked graph autoencoder Bandana via the specialized Boltzmann-Gibbs masking and layer-wise prediction, and thoroughly explore its empirical and theoretical superiority. We demonstrate that Bandana can learn more precise graph manifolds and outperform other baselines, including the state-of-the-art contrastive methods and FeatRecs, on link prediction and the feature-related node classification, solely by pre-training on a structure-learning pretext. While Bandana may not represent the optimal solution, it is the first attempt to explore a new paradigm for masked graph autoencoders that diverges from the discrete mask-then-reconstruct stereotype.

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

# A  MORE THEORETICAL DETAILS

## A.1  Proof of Theorem 3.1

PROOF. It is obvious that the unmasked ego-graph $\mathcal{G}_{i,m}$ has $p(n_i - 1) + 1$ nodes. So

$$
E_D(\mathcal{G}_i) - E_D(\mathcal{G}_{i,m})
$$

$$
= \frac{n_i - 1}{n_i} \| X_i^{\mathcal{G}_i} - X_j^{\mathcal{G}_i} \|^2 - \frac{p(n_i - 1)}{p(n_i - 1) + 1} \| X_i^{\mathcal{G}_i} - X_j^{\mathcal{G}_i} \|^2
$$

$$
= \frac{(n_i - 1)(p(n_i - 1) + 1) - pn_i(n_i - 1)}{pn_i(n_i - 1) + n_i} \| X_i^{\mathcal{G}_i} - X_j^{\mathcal{G}_i} \|^2
$$

$$
= \frac{(n_i - 1)(1 - p)}{pn_i(n_i - 1) + n_i} \| X_i^{\mathcal{G}_i} - X_j^{\mathcal{G}_i} \|^2
$$

$$
\geq 0 \tag{13}
$$

and $E_D(\mathcal{G}_i) - E_D(\mathcal{G}_{i,m}) = 0$ iff $p = 1$. □

## A.2  Proof of Proposition 4.3

PROOF. Under the assumption of $\tilde{T}_j \sim \mathcal{N}(\mu_{\tilde{T}_j}, \Sigma_{\tilde{T}_j})$ and $\Sigma_{\tilde{T}_j} = c\mathbf{I}$, $r_{\mathbf{X}}(\tilde{T}_j)$ is a predictor of the Gaussian mean $\mu_{\tilde{T}_j}$. As such, the negative log likelihood in eq. (8) can be rewritten as an $\ell_2$ error of topological encoding:

$$
\mathcal{L} = -\mathbb{E}_{j \in \mathcal{V}} [\log r_{\mathbf{X}}(\tilde{T}_j)]
$$

$$
= -\mathbb{E}_{j \in \mathcal{V}} \left[ \log \frac{\exp\left( -\frac{1}{2}(r_{\mathbf{X}}(\tilde{T}_j) - \tilde{T}_j)^\top \Sigma_{\tilde{T}_j}^{-1} (r_{\mathbf{X}}(\tilde{T}_j) - \tilde{T}_j) \right)}{(2\pi)^{\frac{n}{2}} \det(\Sigma_{\tilde{T}_j})^{\frac{1}{2}}} \right]
$$

$$
= \frac{1}{2c} \mathbb{E}_{j \in \mathcal{V}} \left[ (r_{\mathbf{X}}(\tilde{T}_j) - \tilde{T}_j)^\top (r_{\mathbf{X}}(\tilde{T}_j) - \tilde{T}_j) \right]
$$

$$
+ \underbrace{\log((2\pi)^{\frac{n}{2}} \det(\Sigma_{\tilde{T}_j})^{\frac{1}{2}})}_{const}
$$

$$
\propto \mathbb{E}_{j \in \mathcal{V}} [\| r_{\mathbf{X}}(\tilde{T}_j) - \tilde{T}_j \|^2] \tag{14}
$$

expanding $r_{\mathbf{X}}(\cdot)$ with the first-order Taylor series yields

$$
r_{\mathbf{X}}(\tilde{T}_j) = r_{\mathbf{X}}(T_j + \epsilon) = r_{\mathbf{X}}(T_j) + \nabla r_{\mathbf{X}}(T_j)\epsilon + o(\epsilon^\top \epsilon) \tag{15}
$$

and we have

$$
\mathcal{L} = \mathbb{E}_{j \in \mathcal{V}} [\| r_{\mathbf{X}}(T_j) + \nabla r_{\mathbf{X}}(T_j)\epsilon - (T_j + \epsilon) + o(\epsilon^\top \epsilon)\|^2]
$$

$$
= \mathbb{E}_{j \in \mathcal{V}} [\| (r_{\mathbf{X}}(T_j) - T_j) + (\nabla r_{\mathbf{X}}(T_j)\epsilon - \epsilon)\|^2] + o(\sigma_\epsilon^2)
$$

$$
= \mathbb{E}_{j \in \mathcal{V}} [\| r_{\mathbf{X}}(T_j) - T_j \|^2]
$$

$$
+ 2\mathbb{E}_{j \in \mathcal{V}}[\epsilon]^\top \mathbb{E}_{j \in \mathcal{V}} [(\nabla r_{\mathbf{X}}(T_j) - \mathbf{I})^\top (r_{\mathbf{X}}(T_j) - T_j)]
$$

$$
+ \left( \mathbb{E}_{j \in \mathcal{V}}[\|\nabla r_{\mathbf{X}}(T_j)\epsilon\|^2] + \mathbb{E}_{j \in \mathcal{V}}[\epsilon^\top \epsilon] \right.
$$

$$
\left. -2\mathbb{E}_{j \in \mathcal{V}}[\epsilon]^\top \mathbb{E}_{j \in \mathcal{V}}[\nabla r_{\mathbf{X}}(T_j)] \right) + o(\sigma_\epsilon^2)
$$

$$
= \mathbb{E}_{j \in \mathcal{V}} [\| r_{\mathbf{X}}(T_j) - T_j \|^2]
$$

$$
+ \text{tr}(\mathbb{E}_{j \in \mathcal{V}}[\epsilon \epsilon^\top] \mathbb{E}_{j \in \mathcal{V}}[\nabla r_{\mathbf{X}}(T_j)^\top \nabla r_{\mathbf{X}}(T_j)])
$$

$$
+ 2\mu_\epsilon^\top \mathbb{E}_{j \in \mathcal{V}} [(\nabla r_{\mathbf{X}}(T_j) - \mathbf{I})^\top (r_{\mathbf{X}}(T_j) - T_j)]
$$

$$
- 2\mu_\epsilon^\top \mathbb{E}_{j \in \mathcal{V}}[\nabla r_{\mathbf{X}}(T_j)] + o(\sigma_\epsilon^2) \tag{16}
$$

As the noise vector of each ego-graph $\mathcal{G}_i$ is a probabilistic simplex, the mean of noises over every edge in $\mathcal{G}_i$ is $1/\deg(i)$. This derives the statistical mean of bandwidths on the entire graph $\mu_\epsilon$ as

$$
\mu_\epsilon = \frac{1}{2|\mathcal{E}|} \sum_{i \in \mathcal{V}} \deg(i) \cdot \frac{1}{\deg(i)} = \frac{n}{2|\mathcal{E}|} \tag{17}
$$

Therefore, $\mu_\epsilon \to 0$ when $n \ll 2|\mathcal{E}|$. In that case,

$$
\mathcal{L} = \mathbb{E}_{j \in \mathcal{V}} [\| r_{\mathbf{X}}(T_j) - T_j \|^2] + \sigma_\epsilon^2 \text{tr}(\mathbb{E}_{j \in \mathcal{V}}[\nabla r_{\mathbf{X}}(T_j)^\top \nabla r_{\mathbf{X}}(T_j)]) + o(\sigma_\epsilon^2)
$$

$$
= \mathbb{E}_{j \in \mathcal{V}} [\| r_{\mathbf{X}}(T_j) - T_j \|^2] + \sigma_\epsilon^2 \mathbb{E}_{j \in \mathcal{V}} [\| \nabla r_{\mathbf{X}}(T_j) \|_F^2] + o(\sigma_\epsilon^2) \tag{18}
$$

□

## A.3  Proof of Theorem 4.4

PROOF. We follow [1] to complete the proof. From a generative perspective, one may consider the edge set of $\mathcal{G}_j$ as a sampled subset from $p(T_j)$. Let

$$
f(T_j, r_{\mathbf{X}}, \nabla r_{\mathbf{X}}) := p(T_j)(\mathbb{E}_{j \in \mathcal{V}} [\| r_{\mathbf{X}}(T_j) - T_j \|^2]
$$

$$
+ \sigma_\epsilon^2 \mathbb{E}_{j \in \mathcal{V}} [\| \nabla r_{\mathbf{X}}(T_j) \|_F^2]) \tag{19}
$$

Then the bandwidth prediction in eq. (11) can be transformed into finding the extremum of an integral functional $\mathcal{L}(r_{\mathbf{X}})$:

$$
r^* = \arg\min \mathcal{L}(r_{\mathbf{X}}), s.t. \ \mathcal{L}(r_{\mathbf{X}}) = \int_{\mathbb{R}^n} f(T_j, r_{\mathbf{X}}, \nabla r_{\mathbf{X}}) \mathrm{d}T_j \tag{20}
$$

Despite a multivariate functional, it can be split into individual components:

$$
\mathcal{L}(r_{\mathbf{X}}) = \sum_{i=1}^n \int_{\mathbb{R}^n} p(T_j) \left( (r_{\mathbf{X},i}(T_j) - T_{ij})^2 + \sigma_\epsilon^2 \sum_{k=1}^n \left( \frac{\partial r_{\mathbf{X},i}(T_j)}{\partial T_{kj}} \right)^2 \right) \mathrm{d}T_j \tag{21}
$$

We know by the Euler-Langrage equation that the optimal $r^*$ satisfies

$$
\left. \frac{\partial f}{\partial r_{\mathbf{X}}} \right|_{r^*} - \frac{\mathrm{d}}{\mathrm{d}T_j} \left. \frac{\partial f}{\partial \nabla r_{\mathbf{X}}} \right|_{r^*} = 0 \tag{22}
$$

By eq. (19), we have

$$
\frac{\partial f}{\partial r_{\mathbf{X}}} = 2(r_{\mathbf{X},i}(T_j) - T_{ij})p(T_j), \tag{23}
$$

$$
\frac{\partial f}{\partial (\nabla r_{\mathbf{X}})_i} = 2\sigma_\epsilon^2 p(T_j) \left[ \frac{\partial r_{\mathbf{X},k}(T_j)}{\partial T_{ij}} \right]_k^\top \tag{24}
$$

$$
\Rightarrow \frac{\partial}{\partial T_{ij}} \frac{\partial f}{\partial (\nabla r_{\mathbf{X}})_i} = 2\sigma_\epsilon^2 \left( \frac{\partial p(T_j)}{\partial T_{ij}} \left[ \frac{\partial r_{\mathbf{X},k}(T_j)}{\partial T_{ij}} \right]_k^\top + p(T_j) \left[ \frac{\partial^2 r_{\mathbf{X},k}(T_j)}{\partial T_{ij}^2} \right]_k^\top \right) \tag{25}
$$

Putting eq. (23) and eq. (25) into eq. (22) yields

$$
r_{\mathbf{X},k}(T_j) - T_{kj} = \frac{\sigma_\epsilon^2}{p(T_j)} \sum_{i=1}^n \left( \frac{\partial p(T_j)}{\partial T_{ij}} \frac{\partial r_{\mathbf{X},k}(T_j)}{\partial T_{ij}} + p(T_j) \frac{\partial^2 r_{\mathbf{X},k}(T_j)}{\partial T_{ij}^2} \right)
$$

$$
= \sigma_\epsilon^2 \sum_{i=1}^n \left( \frac{\partial \log p(T_j)}{\partial T_{ij}} \frac{\partial r_{\mathbf{X},k}(T_j)}{\partial T_{ij}} + \frac{\partial^2 r_{\mathbf{X},k}(T_j)}{\partial T_{ij}^2} \right) \tag{26}
$$

[1] gives an analytical solution of eq. (26) when $\sigma_\epsilon^2 \to 0$:

$$
\left. r_{\mathbf{X},k}^*(T_j) \right|_{\sigma_\epsilon^2 \to 0} = T_{kj} + \sigma_\epsilon^2 \frac{\partial \log p(T_j)}{\partial T_{ij}} + o(\sigma_\epsilon^2) \tag{27}
$$

so the proof concludes:

$$
r_{\mathbf{X}}^*(T_j) - T_j \propto \nabla \log p(T_j) \tag{28}
$$

This indicates that the perturbed topological encoding manifold $p(\tilde{T})$ is approximately equal to the original manifold $p(T)$ when $\epsilon$ is small enough. Hence, despite the changing bandwidths, the

optimizing objective remains invariant as the topological manifold of the original graph data. □

### A.4 Mildness of Assumptions

*A.4.1 The noise mean $\mu_\epsilon$.* Proposition 4.3 and Theorem 4.4 hold under $n \ll 2|\mathcal{E}|$, that is, the mean of bandwidths over every edge needs to be small enough (in other words, the *mask ratio* needs to be close to 1). According to eq. (17), Bandana's mask ratio is a fixed $p = 1 - \mu_\epsilon = 1 - n/2|\mathcal{E}_{train}|$, which is very large in large-scale networks (and even larger in practice because the graph data is not always connected), thus the assumption can be easily satisfied. For discrete TopoRecs, this also implies that little information is available during training. Yet, Bandana keeps the global topology intact, which is conducive to mitigating the impact of the extremely high mask ratio. The mask ratios of Bandana throughout our experiments are listed in Table 4, where "Calculated" represents the mask ratios calculated by $p = 1 - n/2|\mathcal{E}_{train}|$, and "Measured" represents the actual mask ratios measured during training.

*A.4.2 The covariance $\Sigma_{\tilde{T}_j}$.* Another prerequisite of Proposition 4.3 and Theorem 4.4 is that the covariance of $\{\tilde{T}_j\}_{j=1}^n$ should satisfy $\Sigma_{\tilde{T}_j} = c\mathbf{I}$ with an arbitrary constant $c$. It is obvious that $c$ is the variance of noise $\sigma_\epsilon$, so we mainly focus on the diagonal covariance matrix, which implies the independence between every two different entries of $\tilde{T}_j$. As

$$\mathbb{E}[\tilde{T}_{ij}\tilde{T}_{kj}] = \mathbb{E}[(T_{ij} + \epsilon_{ij})(T_{kj} + \epsilon_{kj})]$$
$$= \mathbb{E}[T_{ij}T_{kj} + T_{ij}\epsilon_{kj} + T_{kj}\epsilon_{ij} + \epsilon_{ij}\epsilon_{kj}]$$
$$= \mathbb{E}[T_{ij}T_{kj}] + \mathbb{E}[T_{ij}]\mathbb{E}[\epsilon_{kj}] + \mathbb{E}[T_{kj}]\mathbb{E}[\epsilon_{ij}] + \mathbb{E}[\epsilon_{ij}]\mathbb{E}[\epsilon_{kj}]$$
$$= \mathbb{E}[T_{ij}T_{kj}] + \mathbb{E}[T_{ij} + \epsilon_{ij}]\mathbb{E}[T_{kj} + \epsilon_{kj}] - \mathbb{E}[T_{ij}]\mathbb{E}[T_{kj}]$$
$$= \mathbb{E}[\tilde{T}_{ij}]\mathbb{E}[\tilde{T}_{kj}] + \mathbb{E}[T_{ij}T_{kj}] - \mathbb{E}[T_{ij}]\mathbb{E}[T_{kj}] \quad (29)$$

for any $i, k \in \mathcal{N}_j, i \neq k$, it is equivalent to the independence of the local topology $\{T_{ij}\}_{i \in \mathcal{N}_j}$ of every node $j$, i.e. every two incoming edges of $j$ should be independent. While node relationships in real-world networks are more likely to be correlated, this assumption is introduced for the brevity of the mathematical derivation of Proposition 4.3 and Theorem 4.4. Whether they still hold without this assumption necessitates further mathematical analysis.

## B MORE DISCUSSIONS

In this section, we discuss connections between Bandana and other deep learning models, including graph attention models, score-based models, and energy-based models. Our discussions shed light on the reliability of Bandana from different perspectives, and we hope they will lead to deeper insights in the future.

### B.1 Graph Attention Models

The way we use bandwidth to do weighted message propagation is inspired by the graph self-attention mechanism [4, 35, 67, 76]. Existing studies have pointed out that attention weights should be able to distinguish different edges [13], and softmax-based attention can amplify the dispersion of attention weights to be more discriminative [54]. Therefore, we hold that it is beneficial to generate bandwidth values from a softmax-amplified distribution, i.e. the Boltzmann-Gibbs distribution. Yet, **our bandwidth masking and graph attention mechanisms are fundamentally different**. Graph attention models empirically *fit* a locally optimal weight

**Table 4: Mask ratios of Bandana on various datasets.**

| Dataset | Calculated | Measured |
|---|---|---|
| Cora | 0.6983 | 0.7077 |
| CiteSeer | 0.5702 | 0.6048 |
| PubMed | 0.7383 | 0.7571 |
| Photo | 0.9622 | 0.9630 |
| Computers | 0.9671 | 0.9679 |
| CS | 0.8683 | 0.8697 |
| Physics | 0.9182 | 0.9185 |
| ogbn-arxiv | 0.9140 | 0.9158 |
| ogbl-collab | 0.8840 | 0.8995 |

distribution of neighborhood, in which the parameter matrices converge as training goes on. Our bandwidth masking strategy does not learn the weights, but *randomly generates them parameter-free*. For every iteration, each edge is randomly assigned a different bandwidth, so that different neighbors will be noticed every time to help the encoder distinguish their messages. Plus, the layer-wise masking is inspired by SuperGAT [35] which, however, does not give any explanations, discussions, or even empirical results in terms of the layer-wise approach. Our work also bridges this gap.

Bandana is currently not able to directly pre-train GAT and Graph Transformers as it also assigns weights to every edge in message propagation. It needs to be adjusted to accommodate bandwidths and attention weights. How to improve the topological learning performance of graph attention-based networks in self-supervision now remains an interesting future work.

### B.2 Score-based & Energy-based Models

By Theorem 4.4, bandwidth prediction is equivalent to optimizing the gradient of a log probability $\nabla \log p(T_j)$. This is also called a "*score*" in the field of generation, which can be directly estimated by Score Matching [29, 61] to generate samples that match the original data distribution. Therefore, Bandana can be seen as an implicit score-based model that adds Gaussian noise to the graph topology and learns its score.

Energy-based models (EBMs) [40] perform an alternate optimization process: (i) **optimizing the output**, and (ii) **optimizing the energy**. (i) The forward pass (or inference) of the model $f(x; \Theta) : x \mapsto y$ is viewed as finding the local minimum point $y^*$ on a manifold $E_\Theta \in \mathcal{F}$. Here $E_\Theta : \mathcal{X} \times \mathcal{Y} \rightarrow \mathbb{R}$ is a scoring function of the input-output pair $(x, y)$, judging whether the output $y$ matches $x$ the best. $\mathcal{F}$ is the function space of $E_\Theta$. $E_\Theta(x, y)$ is smaller if $y$ better matches $x$. As the learning process goes on, $y$ has an increasing tendency for minimizing $E_\Theta(x, y)$, and $y = y^*$ when the model converges. Analogous to the principle of minimum energy in thermodynamics, $E_\Theta$ is called an *energy function*. (ii) The backward pass of $f(x, \Theta)$ is viewed as searching on $\mathcal{F}$ for the optimal $E_\Theta$ that meets the above conditions. As the learning process goes on, $E_\Theta(x, y)$ has an increasing tendency to assign lower energy values to more compatible $(x, y)$ pairs and higher values to less compatible ones.

Probabilistic discriminative models $f(x; \Theta) : x \mapsto \hat{y}$ based on maximum likelihood estimation can directly define the energy as

its negative output logit (i.e. the unnormalized probability) because it indicates which $\hat{\mathbf{y}}$ matches $x$ the best. However, as a non-probabilistic model, an autoencoder cannot define the energy in this way. This issue has been solved by [34, 62, 69] who state that the reconstruction of denoising autoencoders is equivalent to performing regularized score matching, and the energy function can be derived using the antiderivative of the score (for any output $\hat{y}$):

$$E_\Theta(X) = \log p(X) = \int (r(X; \Theta) - X)\mathrm{d}X \tag{30}$$

By Proposition 4.3, discrete TopoRecs are not denoising autoencoders and hence not EBMs in this way. On the contrary, Bandana can be viewed as an EBM. It can be inferred from Theorem 4.4 that the manifold of $r_\mathbf{X}(\mathbf{T}) - \mathbf{T}$ is a gradient vector field $\mathcal{T}$, on which inference is allowed to perform gradient descent on $\mathcal{T}$ to find the output $r_\mathbf{X}(\mathbf{T})$ that is closest to the input $\mathbf{T}$. Hence, the following corollary holds:

---

COROLLARY B.1 (Bandana is energy-based). *Let $\mathcal{T} = E_{\mathbf{X},\Theta}(\mathbf{T})$ be an energy landscape similarly defined by eq.* (30). *Then* Bandana's *forward and backward passes are equivalent to implicitly performing the following optimization tasks on $\mathcal{T}$:*

$$\text{Forward pass: } r_\mathbf{X}(\tilde{\mathbf{T}}) = \arg\min_{\tilde{\mathbf{T}}} E_{\mathbf{X},\Theta}(\tilde{\mathbf{T}})$$

$$\text{Backward pass: } E^*_{\mathbf{X},\Theta} = \arg\min_{E \in \mathcal{F}} \mathcal{L}$$

---

Such correlation between Bandana and EBM provides another perspective of the reliability and flexibility of bandwidth mechanisms. We see this as a good foundation for future insights.

## C MORE CONFIGURATIONS

In this section, we detail our experimental configurations to provide reproducibility.

### C.1 Data Statistics

We use a total of 9 real-world datasets. They are categorized and briefly introduced as follows.

*Citation networks.* Cora, CiteSeer, and PubMed [58] are three benchmark datasets often used for semi-supervised node classification. While Cora and CiteSeer consist of research papers mainly in the field of computer science, PubMed is a collection of scientific abstracts from the field of biomedicine and life sciences. Each dataset contains a bag-of-words feature matrix as well as a citation graph, with each node a paper and each edge a citation between two papers. Each paper is associated with a label indicating one of several pre-defined categories, such as Neural Networks, Reinforcement Learning for Cora and Artificial Intelligence (AI), Information Retrieval (IR) for CiteSeer.

*Co-purchase networks.* Photo and Computers [59] are two networks that represent the co-purchase relations of goods in Amazon. Edges represent that two goods are purchased together more frequently. The node features are bag-of-words encoding product reviews. The class labels are given by the categories of products.

*Co-author networks.* CS and Physics [59] are Microsoft Academic Graph (MAG) [71] datasets based on the KDD Cup Challenge in 2016. Each node represents an author of a paper, the keywords of

**Table 5: Dataset statistics.** "†" marks the synthetic ones.

| Dataset | #nodes | #edges | #features | #classes | Density (‰) |
|---|---|---|---|---|---|
| Swiss Roll† | 500 | 6,712 | – | – | 26.9 |
| Two-moon† | 2,000 | 12,264 | – | – | 3.07 |
| Cora | 2,708 | 10,556 | 1,433 | 7 | 1.44 |
| CiteSeer | 3,327 | 9,104 | 3,703 | 6 | 0.82 |
| PubMed | 19,717 | 88,648 | 500 | 3 | 0.23 |
| Photo | 7,487 | 119,043 | 745 | 8 | 4.07 |
| Computers | 13,381 | 245,778 | 767 | 10 | 2.60 |
| CS | 18,333 | 81,894 | 6,805 | 15 | 0.24 |
| Physics | 34,493 | 247,962 | 8,415 | 5 | 0.21 |
| ogbn-arxiv | 169,343 | 2,315,598 | 128 | 40 | 0.08 |
| ogbl-collab | 235,868 | 2,570,930 | 128 | – | 0.05 |

which are encoded as features. Two linked authors have collaborated on a single paper. The meaning of each category tag is the most active research direction of each author.

*OGB networks.* ogbn-arxiv and ogbl-collab [28] are two large-scale undirected networks provided by the Open Graph Benchmark. ogbn-arxiv is a citation network of computer science papers published in arXiv. Features of the papers are obtained by averaging the word embeddings generated by word2vec [48] in their titles and abstracts. In addition, all papers are also associated with the year that the corresponding paper was published. ogbl-collab is a large-scale collaboration network between authors indexed by MAG, with each node representing an author and edge indicating a collaboration between two authors. Features of the authors are the averaged word embeddings of their papers, too. As the original graph is dynamic with the existence of multiple edges between two authors representing the collaborations in different years, we only keep the papers published in 2010 and beyond for training.

Detailed statistics of all datasets are listed in Table 5. "Density" stands for the percentage of all potential connections in a network that are actually positive edges, formally $\rho = \frac{|\mathcal{E}|}{n(n-1)}$.

### C.2 Hardware & Environments

Bandana is built upon PyTorch [53] 1.12.1 and PyTorch Geometric (PyG) [12] 2.3.1. The latter provides all 7 datasets used throughout the quantitative experiments except ogbn-arxiv and ogbl-collab, which are from the OGB 1.3.5 package [28]. Two synthetic datasets used in Section 5.2 come from the PyGSP package [9]. All experiments are conducted on a 24GB NVIDIA GeForce GTX 3090 GPU with CUDA 11.3.

### C.3 Model Setup & Hyperparameters

*Training setup.* We follow the train/validation/test split of previous work [42]. To be specific, we use all existing official splits. For all datasets, edge sets are divided into $\mathcal{E}_\text{train} : \mathcal{E}_\text{val} : \mathcal{E}_\text{test} = 85\% : 5\% : 10\%$ for training and the downstream link prediction. As for node classification, the official split of Planetoid and ogbn-arxiv is adopted and node sets of other datasets are divided into $\mathcal{V}_\text{train} : \mathcal{V}_\text{val} : \mathcal{V}_\text{test} = 10\% : 10\% : 80\%$.

Bandana employs a GCN encoder ($\tilde{\mathbf{G}} = \Sigma_e \hat{\tilde{\mathbf{D}}}^{-1/2} \hat{\tilde{\mathbf{A}}} \hat{\tilde{\mathbf{D}}}^{-1/2}$ in eq. (6)) with 1 to 5 layers and a fixed 2-layer MLP decoder with dropout. For brevity, Bandana does not resort to extra techniques such as path masking, degree regression [42], cross-correlation

**Table 6: Detailed hyperparameters of Bandana.**

| Dataset | Cora | CiteSeer | PubMed | Photo | Computers | CS | Physics | ogbn-arxiv |
|---|---|---|---|---|---|---|---|---|
| No. of layers | 3 | 5 | 2 | 2 | 3 | 2 | 2 | 4 |
| Learning rate $\gamma$ | 1e-2 | 2e-2 | 1e-3 | 2e-3 | 1e-3 | 1e-2 | 2e-3 | 5e-4 |
| Bandwidth temperature $\tau$ | 0.9 | 0.2 | 0.2 | 1 | 0.4 | 1e-6 | 0.4 | 0.4 |
| Intermediate feature dim. | 256 | 256 | 64 | 256 | 256 | 64 | 256 | 256 |
| Output feature dim. | 256 | 256 | 32 | 64 | 64 | 32 | 128 | 256 |
| Encoder dropout | 0.8 | 0.8 | 0.6 | 0.8 | 0.5 | 0.8 | 0.8 | 0.2 |
| Decoder dropout | 0 | 0 | 0.7 | 0.2 | 0.2 | 0.2 | 0.2 | 0 |
| Weight decay (for encoder) | 5e-5 | 5e-5 | 5e-5 | 5e-5 | 0 | 5e-5 | 5e-5 | 5e-5 |
| Weight decay (for linear probing) | 5e-3 | 1e-1 | 5e-5 | 5e-4 | 5e-4 | 1e-3 | 1e-3 | 1e-4 |

**Table 7: Average AUC (%) of link prediction under the end-to-end training/fine-tuning (ETE/FT) and the dot-product probing (DPP). "NODATA" means that the model cannot perform due to the specific data format.**

| Model | Setup | Cora | CiteSeer | PubMed | Photo | Computers | CS | Physics |
|---|---|---|---|---|---|---|---|---|
| T-BGRL [60] | FT | 91.34 | 95.70 | 95.70 | 98.22 | 97.76 | 95.91 | 96.42 |
| | DPP | 73.18 ($\downarrow$**18.2**) | 78.11 ($\downarrow$**17.6**) | 76.21 ($\downarrow$**19.5**) | 80.80 ($\downarrow$**17.9**) | 84.60 ($\downarrow$**13.2**) | 70.08 ($\downarrow$**25.8**) | 89.18 ($\downarrow$7.24) |
| S2GAE [63] | ETE | 93.41 | 93.14 | 98.34 | NODATA | NODATA | NODATA | NODATA |
| | DPP | 89.27 ($\downarrow$4.14) | 86.35 ($\downarrow$6.79) | 89.53 ($\downarrow$8.81) | NODATA | NODATA | NODATA | NODATA |
| MaskGAE-edge [42] | ETE | 96.46 | 97.91 | 98.84 | 98.73 | 98.72 | 98.92 | 95.10 |
| | DPP | 95.66 ($\downarrow$0.80) | 97.02 ($\downarrow$0.89) | 96.51 ($\downarrow$2.33) | 81.12 ($\downarrow$**17.6**) | 76.23 ($\downarrow$**22.5**) | 96.50 ($\downarrow$2.42) | 93.09 ($\downarrow$2.01) |
| MaskGAE-path [42] | ETE | 96.43 | 97.92 | 98.74 | 98.56 | 98.73 | 98.72 | 98.76 |
| | DPP | 95.47 ($\downarrow$0.96) | 97.21 ($\downarrow$0.71) | 97.19 ($\downarrow$1.55) | 80.46 ($\downarrow$**18.1**) | 73.24 ($\downarrow$**25.5**) | 92.26 ($\downarrow$6.46) | 94.00 ($\downarrow$4.76) |
| Bandana | ETE | 95.84 | 97.49 | 97.32 | 97.61 | 96.38 | 98.50 | 98.53 |
| | DPP | 95.71 ($\downarrow$0.13) | 96.89 ($\downarrow$1.08) | 97.26 ($\downarrow$0.06) | 97.24 ($\downarrow$0.37) | 97.33 ($\uparrow$0.95) | 97.42 ($\downarrow$1.08) | 97.02 ($\downarrow$1.51) |

decoding [63], re-masking, and random feature substitution [26]. All non-linear layers (of the encoder, decoder, and learnable downstream branches) are Xavier-initialized with biases 0. Every layer of the encoder is equipped with batch normalization [30], dropout, and an ELU activation function [8]. We perform grid search for the learning rate $\gamma$ and temperature $\tau$ over the searching space {5e-4, 1e-3, 2e-3, 5e-3, 1e-2, 2e-2} and {1e-6, 0.1, 0.2, 0.3, ..., 1} respectively. Adam [36] is used as the model optimizer. For all datasets except ogbn-arxiv, we use a fixed training strategy of 1000 epochs with early stopping, the patience of which is set to 30. For ogbn-arxiv, 100 epochs with batch size $2^{16}$. As with previous work, both grid search and early stopping are carried out on the validation set, and the best validation models are saved for testing.

*Linear probing for node classification.* The so-called linear probing [21] first performs unsupervised pre-training on both the encoder and decoder. Then, the decoder is substituted with a Xavier-initialized linear layer. It is trained (with the encoder frozen) for another 100 epochs with a fixed learning rate of 1e-2 to obtain the classification logits.

All model hyperparameters for node classification are given in Table 6. Please refer to the configuration in our source code for link prediction. They are selected manually on the validation set among several candidate values, except the grid-searched parameters.

# D MORE EXPERIMENT RESULTS

In this section, we showcase the rest of our experimental results to answer some additional research questions:

- ARQ1. *Why is the dot-product probing needed to evaluate self-supervised link prediction?*
- ARQ2. *What is the link prediction performance of* Bandana *on ogbl-collab?*

## D.1 The Dot-product Probing (ARQ1)

In this subsection, we reiterate the necessity of dot-product probing in self-supervised link prediction. We first reveal three shortcomings of the traditional evaluation scheme. Firstly, the end-to-end training has become more like a fully supervised case than self-supervised. Secondly, it is an optimistic evaluation that utilizes the trained parameters on downstream branches, leading to saturation of the link prediction accuracy on many datasets [41]. Lastly, it is unfair to methods that do not learn by link reconstruction. In contrast, the dot-product probing solely takes the latent representations provided by the encoder and directly gets the link prediction results. It does not involve any additional downstream training process, so it enables a more decoupled evaluation about **how the encoder learns, rather than how the encoder-decoder learns**. However, we have observed that the excellent link prediction results of many graph self-supervised models are mainly credited to their decoders, i.e., it is actually their decoders that get well pre-trained.

We consider several baselines specifically designed for link prediction, namely T-BGRL, S2GAE, and MaskGAE. Their link prediction performance is compared by two evaluation strategies: end-to-end training or fine-tuning (ETE/FT), and dot-product probing (DPP). For ETE/FT, T-BGRL fine-tunes with a 1-layer Hadamard-product MLP decoder, while the others are trained end-to-end with a 2-layer MLP decoder. For DPP, we keep the pre-training setup and

**Table 8: Hits@20(%) and Hits@50(%) of link prediction on ogbl-collab.** Best results in each row are in **bold**. "OOM" stands for "Out-Of-Memory" on a 24GB GPU. "†" marks the baselines implemented by us for the current task because they are not officially implemented.

| Metric | Traditional Autoencoder | | Variational Autoencoder | | | | Contrastive Model | | S2GAE [63] | TopoRec | | |
| | GAE† [37] | ARGA† [52] | VGAE† [37] | ARVGA† [52] | SIG-VAE† [19] | SeeGera† [45] | GRACE† [88] | GCA† [89] | | MaskGAE-edge [42] | MaskGAE-path [42] | Bandana |
| --- | --- | --- | --- | --- | --- | --- | --- | --- | --- | --- | --- | --- |
| Hits@20 | 58.93 ± 1.13 | 60.06 ± 1.41 | 45.53 ± 1.87* | 27.32 ± 2.93* | OOM | OOM | OOM | OOM | 40.85 ± 3.30 | 58.59 ± 1.19 | 58.43 ± 1.06 | **60.42 ± 0.84** |
| Hits@50 | 65.97 ± 0.63 | 66.03 ± 0.66 | | | | | | | 54.10 ± 1.16 | 65.58 ± 0.43 | 65.58 ± 0.58 | **67.77 ± 0.72** |

*We obtain much lower scores for VGAE and ARVGA on ogbl-collab than those given by MaskGAE [42]. We report the Hits@50 from [42] instead.

simply replace the decoder with a dot-product decoder during link prediction evaluation. Table 7 shows that, under the DPP setting, the link prediction accuracies of all baselines on a vast majority of datasets show a noticeable decrease compared to those under the ETE/FT setup. As a contrastive method, T-BGRL is more dependent on the fine-tuning process for link prediction, so its performance is the most vulnerable out of all baselines. However, the same phenomenon is observed on the TopoRecs as well. The performance of MaskGAE even decreases by more than 17% and 22% on Photo and Computers, respectively. This suggests that the probing strategy is indispensable for a self-supervised model even if it uses a shared task for both the pretext and the downstream objective. Furthermore, the obvious sensitivity to the DPP setting indicates that the baseline models' excellent prediction accuracy is largely contributed by their "auxiliary" trained decoders. Bandana, however, is

relatively immune to DPP, indicating that its effectiveness is indeed from the pre-trained encoder.

## D.2 Link Prediction on ogbl-collab (ARQ2)

Here we provide the additional link prediction results on ogbl-collab of Bandana and the baselines we have reproduced and implemented. We keep the end-to-end training on ogbl-collab (on which we observe that almost all models fail with the dot-product probing, as it may be too hard to achieve on large-scale datasets). We report Hits@20 and Hits@50 in Table 8. Bandana outperforms both configurations of MaskGAE by about 2%, indicating its advantage of topological learning on large-scale data.

Received 20 February 2007; revised 12 March 2009; accepted 5 June 2009

