# OpenReview forum: "Masked Graph Autoencoder with Non-discrete Bandwidths"
_ACM.org/TheWebConf/2024/Conference — TheWebConf24_

### Official Review · Reviewer_sz9j · 2023-11-22

**Novelty:** 5
**Technical Quality:** 5

**Review:**

The paper introduces Bandana, a novel masked graph autoencoder framework. Bandana diverges from the traditional discrete mask-then-reconstruct approach, employing a bandwidth masking and reconstruction scheme. It uses non-discrete edge masks sampled from a continuous and dispersive probability distribution, aiming to overcome limitations like blocking message flows and suboptimal neighborhood discriminability inherent in binary link reconstruction strategies. The framework's effectiveness is demonstrated both theoretically and empirically, outperforming representative baselines in link prediction and node classification tasks.


[+] The introduction of non-discrete masking and bandwidth prediction is a significant departure from traditional methods, potentially offering more nuanced and efficient graph representation learning.

[+] The paper provides both theoretical insights and empirical evidence to support the superiority of the Bandana framework over existing methods.

[+] Extensive experiments across multiple datasets validate the model's efficacy and robustness.

[-] Better to give more analysis about how bandwidths affect the results. More case studies would be better.

[-] Better to run larger datasets.

**Questions:**

1. How does the Bandana framework manage the computational complexity introduced by non-discrete bandwidths?
2. Is there a risk of overfitting with the Bandana model, especially in scenarios with limited training data?
3. Can the authors provide more insights into the interpretability of the model, especially regarding how bandwidths affect message propagation?

**Reviewer Confidence:**

2: The reviewer is willing to defend the evaluation, but it is likely that the reviewer did not understand parts of the paper

**Scope:**

3: The work is somewhat relevant to the Web and to the track, and is of narrow interest to a sub-community

---

### Official Review · Reviewer_h9UR · 2023-11-22

**Novelty:** 5
**Technical Quality:** 6

**Review:**

In this paper, the authors propose a new masked graph autoencoder method for graph self-supervised learning. Specifically, the authors find that the existing discrete edge mask and binary link reconstruction methods suffer from block message flows, over-smoothing, and suboptimal neighborhood discriminability. To tackle these issues, the authors propose Bandana, a non-discrete edge mask method using bandwidth masking and layerwise bandwidth prediction. Experimental results on node classification and link prediction demonstrate the effectiveness of the proposed method.

Pros:
(1) Self-supervised learning on graphs is a trending direction and masked graph autoencoder is a promising solution.
(2) The paper establishes a theoretical relationship between the bandwidth mechanism and regularized denoising autoencoders, providing a solid theoretical basis for its approach.
(3) The authors have provided the source codes as well as experimental details for reproducibility.
(4) The paper is well-written in general.

Cons and questions:
(1) The improvement of the proposed method over GraphMAE for node classification is somewhat marginal.
(2) There is no discussion regarding the time complexity or efficiency of the proposed method, which could be added.
(3) I also wonder whether the proposed approach can be generalized to different types of graphs beyond those tested in the paper?

**Questions:**

See above

**Ethics Review Description:**

N.A.

**Reviewer Confidence:**

3: The reviewer is confident but not certain that the evaluation is correct

**Scope:**

3: The work is somewhat relevant to the Web and to the track, and is of narrow interest to a sub-community

---

### Official Review · Reviewer_UNcB · 2023-11-24

**Novelty:** 4
**Technical Quality:** 2

**Review:**

## Summary: ##
The paper explores an approach in graph self-supervised learning by addressing the limitations of existing discrete edge masking and binary link reconstruction strategies. The authors propose a topological masked graph autoencoder using bandwidth masking and a layer-wise bandwidth prediction objective. They also demonstrate good graph topological learning ability in the experiments.

## Strengths: ##
- The research topic of masked graph autoencoder is very interesting and important for the graph machine learning community.
- The authors propose solid theoretical analyses to the method.
- The experiments show the effectiveness of the method.

## Weaknesses: ##
- The authors should conduct fair comparasions under the same hyperparameter setting strategy rather than only searching hyperparameter for the proposed method (Table 6).
- More analyses should be present for the experimental results to support the main points in the paper.
- The relations between the present theories and the graph itself seem to be weak. The authors should pay more attention to the properties of graphs instead of general machine learning.
- The experiments should be conducted on more large-scale benchmarks such as more datasets in Open Graph Benchmark besides ogb-arxiv.

**Questions:**

What is the meaning of NODATA (the model cannot perform due to the specific data format) in the tables?

Why does these results cannot be reported?

**Reviewer Confidence:**

4: The reviewer is certain that the evaluation is correct and very familiar with the relevant literature

**Scope:**

3: The work is somewhat relevant to the Web and to the track, and is of narrow interest to a sub-community

---

### Official Review · Reviewer_itKW · 2023-11-28

**Novelty:** 6
**Technical Quality:** 6

**Review:**

Summary:

This paper revisits the problem of graph autoencoding via masking.  It first shows theoretically that removal of random edges to perform masking results in oversmoothed representations.  It goes further, showing *why* this happens.  To overcome this and other shortcomings, the authors propose _Bandana_, a novel masking strategy that samples masks from a continuous distribution.  In the encoder, message passing through edges is _partially_ impeded by the selected bandwidths, and the goal of the decoder is to estimate how much each edge was masked.

Overall, I lean toward acceptance of this paper.

Pros:

1.) The theory and experiments complement one another well.

2.) The performance of Bandana on a variety of tasks is a substantial empirical improvement on state of the art methods.

3.) The writing is clear.

**Questions:**

N/A

**Reviewer Confidence:**

3: The reviewer is confident but not certain that the evaluation is correct

**Scope:**

4: The work is relevant to the Web and to the track, and is of broad interest to the community

---

### Official Review · Reviewer_pid2 · 2023-12-01

**Novelty:** 6
**Technical Quality:** 6

**Review:**

The paper proposes a new masked graph autoencoder model with structure-aware non-discrete bandwidths. The major idea is to change discrete masks to non-discrete edge masks with a continuous and dispersive probability distribution. The problem studied in this work is popular and important. Experiments on several datasets (from small to large) demonstrate that the proposed model outperforms many baseline methods. Additional analyses such as ablation and parameter analysis are provided. Overall, this is a good work.

Strength

+ Motivation: The problem studied in this work is popular and important. The motivation is clear.

+ Method: The proposed method is novel. It is interesting and new to explore non-discrete edge masks. Theoretical justifications are also provided.

+ Experiments: Experiments are extensive and the proposed method outperforms baseline methods.

+ Presentation: The paper writing is good and well organized.

Weakness/questions

-  I checked the results on baseline methods in their original papers. Their results (e.g., MaskMAE) are better than the results reported in this paper. If using different experiment settings, please explain why and clearly describe the experimental difference. More analyses such as embedding visualization and case studies are suggested.

**Questions:**

See the questions that I mentioned in the above review.

**Reviewer Confidence:**

4: The reviewer is certain that the evaluation is correct and very familiar with the relevant literature

**Scope:**

4: The work is relevant to the Web and to the track, and is of broad interest to the community

---

### Decision · Program_Chairs · 2024-01-22

**Decision:**

Accept

**Comment:**

The paper develops a new self-supervised learning method that uses non-discrete edge masking.

 Pros:
 * Self-supervised learning is an important problem in graph learning and the model trained with the self-supervised method has good performance in both node classification and link prediction.
 * The method is quite novel.
 * The paper is well-written.
 * The paper has extensive evaluation of the method.

 Cons:
 * The datasets used in the paper are quite small. Although the authors show performance numbers on relatively larger datasets, such as ogbn-mag and ogbl-ppa, the authors should use larger datasets in the final version.
 * The evaluation method for link prediction raised concerns from a reviewer. I agree with the reviewer that the evaluation method used in the paper is relatively less common. The authors should consider having more explanations why the evaluation method is used in the paper. If the space allows, maybe the results of both evaluation methods can be provided in the paper.
 * The paper should have analysis of the computation and memory complexity to indicate that the method is scalable to large datasets.